# The role of urban trees in reducing land surface temperatures in European cities

Jonas Schwaab [1✉], Ronny Meier [1], Gianluca Mussetti [1], Sonia Seneviratne [1], Christine Bürgi[1] &
Edouard L. Davin [1,2]

Urban trees influence temperatures in cities. However, their effectiveness at mitigating urban heat in different climatic contexts and in comparison to treeless urban green spaces has not yet been sufficiently explored. Here, we use high-resolution satellite land surface temperatures (LSTs) and land-cover data from 293 European cities to infer the potential of urban trees to reduce LSTs. We show that urban trees exhibit lower temperatures than urban fabric across most European cities in summer and during hot extremes. Compared to continuous urban fabric, LSTs observed for urban trees are on average 0-4 K lower in Southern European regions and 8-12 K lower in Central Europe. Treeless urban green spaces are overall less effective in reducing LSTs, and their cooling effect is approximately 2-4 times lower than the cooling induced by urban trees. By revealing continental-scale patterns in the effect of trees and treeless green spaces on urban LST our results highlight the importance of considering and further investigating the climate-dependent effectiveness of heat mitigation measures in cities.

---

[1] Institute for Atmospheric and Climate Science, ETH Zurich, Zurich, Switzerland. [2] Present address: Wyss Academy for Nature, Climate and Environmental Physics, Oeschger Centre for Climate Change Research, University of Bern, Bern, Switzerland. ✉email: jonasschwaab@ethz.ch

Urban trees can mitigate heat in urban areas and its adverse impacts on human health, energy consumption and urban infrastructure[1,2]. Based on observations, the magnitude by which urban trees and other urban vegetation may reduce urban heat has hardly been systematically assessed for different climatic conditions.

By relying on surface urban heat island (SUHI) data and adopting an energy-balance-based modelling approach, it has been shown that the cooling effect of an increased amount of urban vegetation in tropical cities will be limited and generally differs between wet and dry climates[2]. Since SUHIs are usually estimated as the differences in land surface temperature (LST) between cities and their surroundings, it can be difficult to distinguish the effects of different types of vegetation (e.g. urban trees vs. treeless urban green spaces) on temperature[3,4]. Such a distinction can be crucial, which has been shown in several studies investigating the climatic impacts of land-use/land-cover (LULC) changes[5,6]. These studies look at the effect of different LULC types but do not focus on the urban environment and hence miss regional differences in the potential effects of urban trees and treeless urban green spaces on temperatures. Studies that explicitly focus on different types of LULC in urban contexts often focus on a specific region[7] but do not analyse how different LULC types affect temperatures in different regions.

Trees influence urban climate primarily via shading and transpiration[8] and also via albedo. Shading can strongly reduce daytime LSTs and air temperatures[9], with the effect usually being larger over asphalt than over grass surfaces[10,11] and being larger in shallow than in deep street canyons[12]. The shading effect depends, among other factors, mainly on the morphological characteristics of different trees/tree species and has been shown to increase with leaf area index (LAI)[11,13]. The amount of transpiration and its effect on temperatures depends on the characteristics of trees/tree species but is also strongly dependent on environmental conditions that have, for example, an influence on the stomatal conductance of trees[8]. The environmental conditions that influence the transpiration of trees and their potential to reduce temperatures shift, for instance, with different seasons, during extreme conditions, in different geographical contexts and along gradients of urbanization[14–16].

Seasonality has a strong influence on the cooling potential of trees and vegetation in general[15,17]. In many regions, temperature differences between vegetation and urban fabric are greater during summer than during winter[15]. However, in dry regions including parts of Southern Europe, the summertime cooling provided by vegetation can be reduced due to soil moisture limited evapotranspiration (ET)[17]. As results for the US show, the cooling provided by urban trees during cold extremes is much smaller than during heat extremes, and the amount of transpiration may be closely connected to the variation in saturation vapour pressure[14]. The two opposing effects of increased surface resistance during hot extremes (due to soil moisture limitations and stomatal behaviour) and an increased vapour pressure deficit (VPD; mainly due to increased temperatures) can either lead to an increase or a decrease in temperatures over vegetation during heatwaves[18]. However, our understanding of how temperatures respond to these contrasting effects in different geographical and climatic contexts remains limited.

The potential of trees to reduce temperatures via transpiration is influenced by local- and micro-scale climatic conditions and may differ for trees in a city and trees or forests in rural areas[19,20]. The environmental conditions in an urban context could either increase or decrease the temperature reduction caused by trees[8]. For example, higher $CO_2$ concentrations[21], greater nutrient availability[22], higher temperatures[23] and higher levels of irrigation[24] may regularly be encountered in cities and can

increase transpiration and cooling[25]. On the other hand, several factors that may negatively affect growth of trees and their cooling effect need to be taken into account[26]. High temperatures in cities can increase water stress[27]. Insufficient soil volumes and soil compaction can limit root growth[28] and increased air pollution can have several adverse effects[26]. Due to the different environmental conditions and tree species in cities, it is not clear whether studying rural forests allow us to draw conclusions on the cooling potential of trees within urban areas.

Based on a unique high-resolution data set of remote-sensing based LST (120,285 Landsat scenes) and LULC data of 293 European cities, we compare the temperature differences between urban trees, treeless urban green spaces and urban fabric. In addition, we calculate temperature differences between rural pastures, rural forests and urban fabric (lower LSTs of vegetated areas in comparison to urban areas, i.e. negative temperature differences, are henceforth referred to as cooling). To compare LST differences between these LULC types in different cities, we calibrate Generalized Additive Models (GAMs) for each city and LST observation. These models include the LULC fraction as a predictor variable and allow us to make predictions of the temperature differences between areas that are covered 100% by urban trees, continuous urban fabric or any other land-cover. This allows for a daytime comparison of LST differences among different LULC types at approximately 10:15 a.m. (approximate Landsat acquisition time over Europe). In addition, we separate the effect of different LULC types on temperatures for different conditions (i.e. moderate temperatures vs. hot extremes, see 'Methods' section).

## Results

**LST differences.** During hot temperature extremes, the results indicate a clear difference in LSTs between areas of continuous urban fabric and areas covered by urban trees (Fig. 1). Urban trees are found to have lower temperatures than urban fabric in all analysed European cities with the exception of a few cities in southern Turkey, the Mediterranean and the Iberian Peninsula (e.g. Gaziantep, Fig. 2c). The LST difference is especially high in cities in Central Europe, including the regions of France, Alps/Mid-Europe, British Isles and Eastern Europe, (−12 to −8 K) but lower in the Mediterranean, Turkey and the Iberian Peninsula (−4 to 0 K).

The median summertime temperature difference between urban trees and urban fabric is not always consistent with the temperature difference during hot extremes (Fig. 2c and Supplementary Fig. 1). For instance, the temperature differences during hot extremes in Turkey, the Mediterranean, the Iberian Peninsula, France and Eastern Europe are lower than during average summertime conditions, indicating that the cooling provided by trees decreases during hot extremes in these regions. In contrast, in Scandinavia, the British Isles and parts of the Alps/Mid-Europe, the cooling provided during hot extremes is at times even higher than median summertime cooling. The highest cooling is observed to move further north during hot extremes in comparison to average summertime conditions (Supplementary Fig. 17).

The cooling during different seasons also shows a clear regional pattern (Fig. 2c). In Southern European and Turkish cities such as Gaziantep (Turkey), Cordoba (Spain) and Antalya (Turkey), the cooling during spring (March/April/May) is higher than or very close to the cooling during summer (June/July/August). In European cities in all other regions (cf. Fig. 2c), the cooling is highest during summer. The cooling during autumn (September/October/November) is lowest in all cities and regions in comparison to the cooling in summer and spring.

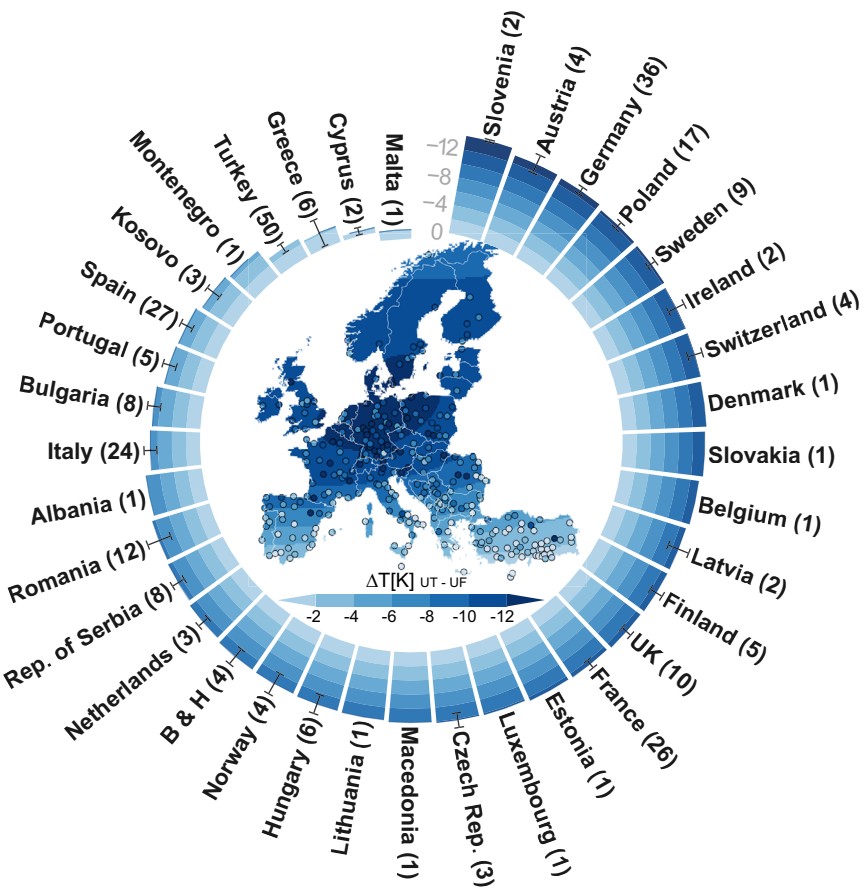

**Fig. 1 Regional variation in temperature differences (ΔT) during hot extremes between areas covered 100% by urban trees (UT) and areas covered 100% by continuous urban fabric (UF).** The map shows smoothed spatial trends of the temperature differences, and each dot represents the temperature difference in a specific city. Mean temperature differences for each country are indicated by stacked bars, and standard errors of the mean are also shown. The number of cities that could be used to calculate mean values in each country are indicated in brackets after the country name. B & H stands for the country Bosnia and Herzegovina.

The temperature differences between rural forests and continuous urban fabric closely resemble the temperature differences between urban trees and urban fabric (Fig. 3, Supplementary Fig. 2, and Supplementary Fig. 3). However, there are some notable distinctions. Urban trees reduce LSTs more than rural forests in Central European regions. In contrast, in Turkey, the reduction in the LSTs of rural forests is larger than that of urban trees. The temperature differences between rural forests and urban fabric ($\Delta T_{\text{F-UF}}$) show an east–west gradient, with the absolute $\Delta T_{\text{F-UF}}$ in Eastern Europe being lower than the temperature differences in Western Europe. Absolute temperature differences between treeless green spaces and urban fabric ($\Delta T_{\text{GS-UF}}$) are smaller than the temperature differences between urban trees and urban fabric ($\Delta T_{\text{UT-UF}}$) in all European regions (Supplementary Fig. 1a). Similarly, the absolute temperature differences between rural pastures and urban fabric ($\Delta T_{\text{P-UF}}$) are much smaller than the ones between rural forests and urban fabric (Supplementary Fig. 1b). Green spaces and pastures are often warmer than urban fabric in Southern European regions and particularly in Turkey. The temperature differences between urban trees and green spaces ($\Delta T_{\text{UT-GS}}$) and rural forests and pastures ($\Delta T_{\text{F-P}}$) show a less clear regional pattern and differ from each other. $\Delta T_{\text{UT-GS}}$ is slightly higher in Central European regions than in Southern European regions, whereas $\Delta T_{\text{F-P}}$ is highest in the Mediterranean and Turkey. ET and albedo also show distinct regional patterns. ET in Southern European regions (particularly in the Iberian Peninsula and Turkey) over forests

and pastures is much lower than in most central European regions (Supplementary Fig. 5). The albedo of urban areas is highest in southern European regions (particularly in the regions Mediterranean and Turkey) and lowest over Scandinavia (Supplementary Figs. 4 and 19). The variation in the albedo of forest areas is relatively small in comparison to the variation in the albedo of urban areas (Supplementary Figs. 4 and 19). This is why the regional differences in albedo between urban and forested areas are consistent with the regional variation in the albedo of urban areas. The inter-city spatial variation in $\Delta T_{\text{UT-UF}}$ in Europe is correlated with the spatial variation in ET over rural forest areas (Fig. 4a). The inter-city variation in $\Delta T_{\text{GS-UF}}$ in Europe is correlated with the spatial variation in ET over rural pastures (Fig. 4b). In contrast, the inter-city correlation between $\Delta T_{\text{UT-UF}}$ and the albedo difference between forested and urban areas ($\alpha_{\text{F-U}}$) is very small, and the inter-city variance of $\Delta T_{\text{UT-UF}}$ can hardly be explained by albedo differences ($R^2 < 0.1$).

**Urban trees for mitigating urban heat in Europe**. Based on observations for a large number of cities in different climates, we compare temperatures over areas of urban trees, treeless urban green spaces, rural forests, rural pastures and continuous urban fabric. The results show that the local cooling of urban trees in comparison to urban fabric varies with background climate. The absolute LST differences between urban fabric and urban trees are the largest in Central Europe pointing towards a high cooling

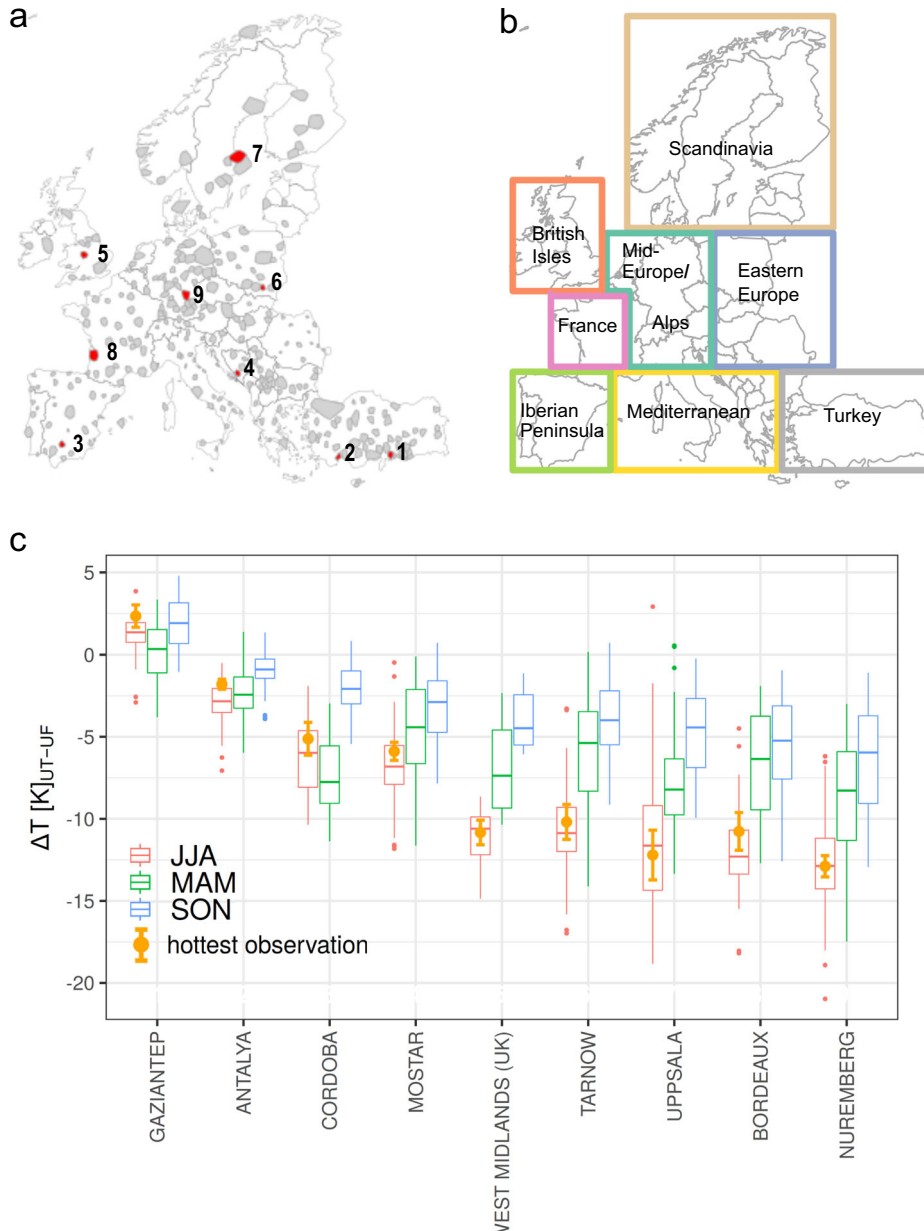

**Fig. 2 Temperature differences between urban trees and continuous urban fabric for selected cities in Europe. a** All cities together with their surroundings that were selected for analysis (grey) and cities for which results are shown in more detail (red). In each region, a representative city was selected (except for Turkey, where we show the results for two cities). **b** Geographic extent of the defined European regions. **c** The LST differences between continuous urban fabric and areas covered 100% by urban trees (UT urban trees, UF continuous urban fabric). Boxplots of each city indicate the spread of temperature differences calculated for all summertime (JJA) observations (boxes show the first and third quartile; whiskers show the largest/smallest values, but do not extend beyond 1.5 times of the interquartile range; outliers are shown as separate points). The temperature difference observed when the background temperature was highest is shown as an orange dot together with error bars denoting standard errors.

potential. The absolute temperature differences between urban trees and urban green spaces ($\Delta T_{UT\text{-}GS}$) are generally much lower than the LST differences between urban trees and urban fabric. However, in several cities in Southern Europe (in the Iberian Peninsula, the Mediterranean and Turkey), the absolute value of $\Delta T_{UT\text{-}GS}$ is small and may even be larger than the absolute difference between urban trees and urban fabric. Hence, it can be argued that it is crucial to equip green spaces with more trees, particularly in these cities and regions. In Scandinavia and over the British Isles, treeless urban green spaces and urban trees provide for substantial cooling. However, while our analysis based

on remote-sensing data shows clear spatial patterns in the LST reduction provided by different vegetation types, it is essential to note that the benefits of trees and green spaces are manifold. For example, pedestrian thermal comfort can substantially vary if the effects of trees on shading, wind speed and humidity are taken into account[12,20,29].

Our results show that the average LST differences between vegetated and urban land can diverge from LST differences during hot extremes. Depending on the region, the differences during hot extremes can be either larger or smaller than those during average summertime conditions. These findings suggest

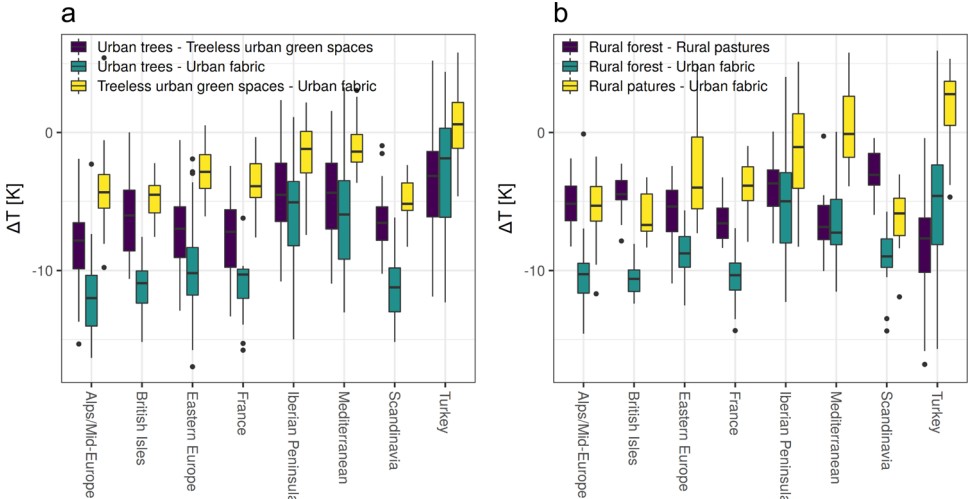

**Fig. 3 Temperature differences between urban or rural vegetation and urban fabric. a** Temperature differences between urban vegetation and urban fabric. **b** Temperature differences between rural vegetation and urban fabric (boxes show the first and third quartile; whiskers show the largest/smallest values but do not extend beyond 1.5 times of the interquartile range; outliers are shown as separate points).

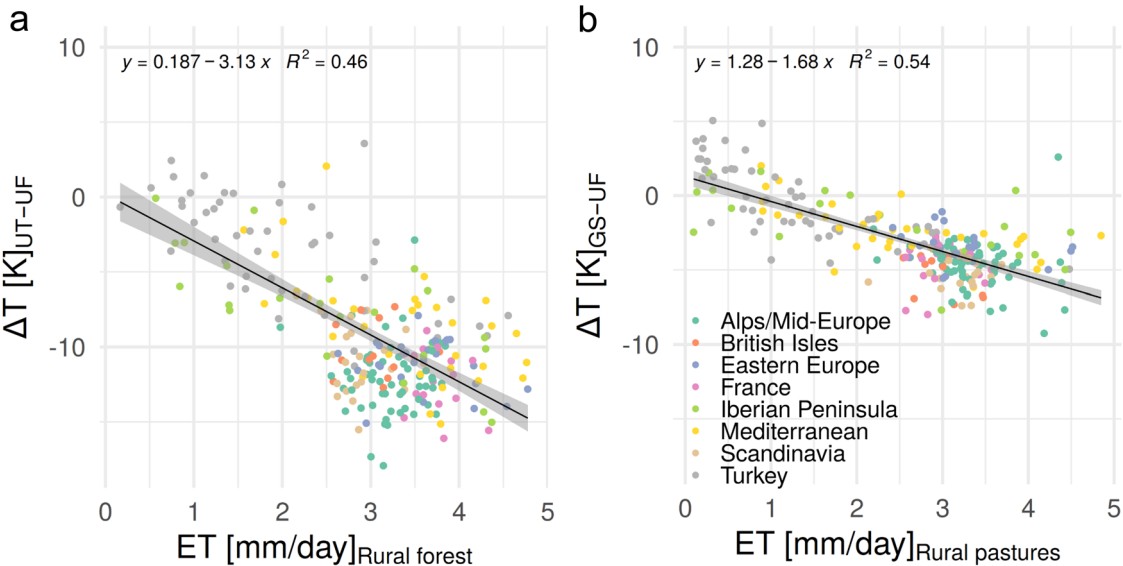

**Fig. 4 Mean summertime temperature differences (ΔT) between urban vegetated areas and continuous urban fabric plotted against evapotranspiration of vegetated areas outside of each city. a** Scatterplot of temperature differences between urban trees (UT) and urban fabric (UF) plotted against evapotranspiration (ET) estimated for rural forests. **b** Scatterplot of temperature differences between treeless urban green spaces (GS) and urban fabric (UF) plotted against evapotranspiration estimated for rural pastures. Each dot represents a city. All cities in a specific region have the same colour.

that temporally averaging LST observations before deriving the impacts of vegetation on temperature may obscure the cooling potential during times it is most important (i.e. during hot extremes). Determining whether high cooling during a short hot period is more relevant than high cooling during longer less extreme periods, therefore, becomes a pertinent component of mitigating the adverse effects of urban heat. In particular, this could be relevant when comparing different heat mitigation strategies that may also have a greater or lesser effect during hot extremes.

The cooling potential of urban trees decreases during hot extremes in many cities, in particular in Southern and Southeastern European regions. Projected drying in European summers in these regions is likely to further reduce vegetation benefits[30]. However, drying may not only occur in Southern Europe but in many European regions[30]. Hence, we may see a decrease in cooling even in regions where we presently see the highest

cooling. Irrigation could help to maintain the high cooling provided by vegetation in these regions but may be limited by future water scarcity. This sheds light on additional heat mitigation measures (e.g. increasing the albedo of roofs and pavements) and shows how difficult it is to compare the effect of different measures for varying environmental conditions.

**Biophysical processes related to observed cooling patterns and differences between urban and rural vegetation**. The temperature differences between urban trees and continuous urban fabric are correlated with the temperature differences between continuous urban fabric and rural forests (Supplementary Fig. 2) and show very similar regional variations (Fig. 3). This close correlation indicates that the cooling provided by urban trees and rural forests in a specific region is guided by similar processes and environmental conditions. In particular, the spatial patterns of temperature differences between urban trees/rural forest and

continuous urban fabric are closely linked to the level of ET associated with forests in different regions. Similarly, the spatial patterns of temperature differences between treeless urban green spaces/rural pastures and urban fabric are quite closely correlated with ET over pastures in the surrounding of each city. ET over vegetated areas explains a large part of the variation in LST differences.

The variation in environmental conditions along urban-to-rural gradients, which can be very important[8], has according to our results a much smaller impact on the variation in cooling than the variation in environmental conditions across regions. However, several differences between rural and urban vegetation cooling are noteworthy (Fig. 3). The cooling of urban trees in Central European regions and, particularly in Scandinavia, is higher than that of rural forests. This could indicate that factors potentially contributing to a higher transpiration and cooling rate in cities (e.g. higher background temperatures) outweigh factors that may reduce cooling in cities (e.g. increasing water stress due to insufficient soil volumes). In Turkey, the cooling of urban trees is generally much lower than that provided by rural forests and hence factors reducing the cooling of urban trees in cities may dominate in this region. On the other hand, the cooling of treeless green spaces in Turkey is higher than that of rural pastures. This could indicate that irrigation of treeless urban green spaces is more relevant than irrigation of urban trees in Southern European regions, including Turkey. Irrigation may, indeed, play a relatively small role for urban trees in Europe[31,32]. However, such aspects need further investigation, and it still is very difficult to derive a clear picture of urban vs. rural vegetation temperature and transpiration differences. To further validate and elucidate the urban vs. rural differences in cooling provided by vegetation, it will be crucial to generate high spatial resolution data on the biophysical processes within cities including e.g. estimates of sensible and latent heat fluxes[33].

The lowest temperature differences between urban trees and urban fabric are observed in cities in Southern European regions and are related to low ET rates (Fig. 4), which can be linked to increased surface resistance due to limited soil moisture availability[18,34]. High temperatures during summertime in the Mediterranean and during hot extremes have the potential to increase ET through the high VPD[16,18]. However, transpirational cooling of trees often decreases considerably due to reduced stomatal conductance[35]. Certain tree species keep their stomata open even during hot extremes, possibly to create a cooling effect through transpiration[36]. Hence, there are regions in which trees show an increase in transpiration during hot extremes[37]. The species-specific response to high temperatures and drought conditions[38] overlays the effect of environmental conditions (e.g. amount of soil moisture) in ways that are not directly captured in the MODIS ET product used in this study and cannot easily be disentangled. Since the cooling of urban trees during hot extremes shifts north and increases over the British Isles, Scandinavia and parts of Mid-Europe/Alps, we assume that higher VPD in combination with sufficient soil moisture availability causes an increase in transpiration in those regions. The decreased cooling during hot extremes in the Mediterranean and Turkey indicates that increased VPD will not lead to a further increase in transpiration in southern regions due to limited soil moisture.

In comparison to ET, albedo plays a minor role in explaining the inter-city temperature differences between urban trees and urban fabric. However, while inter-city differences may not be strongly influenced by albedo, the temperature differences between urban trees and urban fabric in specific regions most likely are. In particular, the albedo can have a larger effect in dryer areas such as Southern Europe[39], and it may increase during hot extremes that are associated with large amounts of incoming shortwave radiation[40]. It is notable that LSTs may be even higher over urban trees than over continuous urban areas in Southern European regions and Turkey (e.g. in Gaziantep). This may be related to extremely low levels of ET over urban tree areas and hence a more significant influence of the high albedo of urban areas in Southern Europe. Lower LAIs in Mediterranean regions[41] could be an additional factor to be considered. If satellites observe a large fraction of dry and even bare soil underneath trees with low LAIs, LSTs may appear to be very high.

There are substantial temperature differences between tree-covered areas and green spaces and between rural forests and rural pastures in several parts of Europe. As a recent study shows, such LST differences are related to high rates of ET being linked to high LAIs of tree-covered areas[42] and hence the study concludes, in accordance with our results, that not only the amount of green spaces but also the type of vegetation exerts a strong control on LSTs and SUHIs. Differences in ET between vegetation types may not only be related to varying LAIs but also to additional physiological and biological characteristics of different vegetation types and their control on ET and surface roughness[6,43–45]. For example, trees are associated with a larger root depth[46] that allows higher exploitation of soil moisture, sustaining larger ET rates when the upper soil layers are dry[44]. Rural trees and forests typically exhibit a high surface roughness, which increases the efficiency of heat convection and may, therefore, also be an important factor explaining the significant temperature differences between rural forests and rural pastures in Southern European regions[47]. For large patches of urban trees and treeless urban green spaces, similar roughness effects as for their rural counterparts (i.e. rural forests and rural pastures) may be relevant. However, the surface roughness of vegetated areas usually interacts in complex ways with the surrounding urban structure. Trees within street canyons can decrease the roughness, leading to reduced turbulent exchange, particularly if trees are smaller than surrounding buildings[16]. If they are higher, they can also increase roughness[48]. Roughness effects may also be important for an explanation of the urban heat island magnitude in different regions since the surrounding of urban areas may convect heat more (wet climates) or less (dry climates) efficiently than urban areas[49]. However, more recent results suggest that the effect of aerodynamic resistance (mainly controlled by surface roughness) is less relevant in explaining the spatial variation of urban heat islands than the imperviousness that controls ET[50].

## Discussion

Our analysis of remote-sensing based LST profits from high spatial resolution and geographic coverage but is limited by temporal resolution. A low temporal resolution and early observation time (around 10:15 a.m.) leads to increased uncertainties particularly when predicting LULC temperature differences during hot extremes for which ideally highly resolved temporal data should be used[51]. In addition, remote sensing LST data is mainly derived during cloud-free conditions[52], and hence it is rather impossible to infer LST differences between vegetated land and urban fabric during cloudy conditions.

The presented results help us to understand large-scale patterns of LST differences, but the results of single cities, as well as absolute temperature differences between urban and vegetated areas, should be interpreted with care. It also has to be noted that, even though our approach seeks to reduce the effect of spatial arrangement of different LULC types on temperature, it is unlikely that we have fully eliminated these effects. The effects of different LULC configurations may also be included directly, for example, by using landscape metrics[53,54]. In addition, the analysis

of Local Climate Zones is an important approach to comprehensively analyse urban areas of mixed LULC[55] and may be complementary to our approach that aims at separating the effect of different LULC types.

Individual or scattered urban trees and thin strips of trees are usually not included in the European urban tree data set that we are using (Supplementary Figs. 6 and 7). Isolated street trees interact differently with the surrounding environment compared to grouped trees[29]. These effects are not fully captured in our analysis and only more precise tree data sets will allow us to better take them into account. It should also be noted that the overall amount of urban vegetation varies between cities in different European regions (Supplementary Fig. 20). The amount of vegetation influences urban environmental variables like temperature and humidity and hence may influence e.g. transpiration of trees and the LSTs that we observe.

Other factors that influence the temperature differences in specific cities are urban morphology and potential anisotropy effects that influence and may bias observed LSTs[56]. For example, the canyon aspect ratio (i.e. the ratio of building height and street width) can have a strong influence on observed temperature differences between urban trees and urban fabric[12]. Potential effects of urban morphology are not directly included in our analysis. However, we may capture some of these effects, because the different urban land-cover categories that we include (e.g. Continuous Urban Fabric or Discontinuous Low Density Urban Fabric) are closely related to the building height (Supplementary Fig. 15) and it shows that including building height as an additional variable does not substantially change our results in selected cities (Supplementary Fig. 16 and Supplementary Note 5). In general, morphological effects may be particularly relevant when comparing the LST reductions related to trees at different locations within each city, but they may have less of an effect when comparing the LST reductions of trees between different cities.

Our analysis is focused on LST, which is less directly related to the adverse impacts of urban heat than air temperatures. The relationship between LST derived from satellite observations and air temperature ($T_a$) is complex[57–59]. Under cloud-free and low wind speed conditions in summer, daily maximum LSTs are usually several degrees higher than air temperatures[60,61]. This is particularly the case over agricultural and barren land but less so over forested land[62]. Accordingly, it has been shown that differences in $T_a$ between forests and grassland are smaller than differences in LST between these two land-covers[63]. Likewise, the SUHI, being based on LST estimates, is often higher than the canopy urban heat island, which is based on $T_a$ estimates[60,64]. While there are clear systematic differences between LST and $T_a$, there are also clear correlations between the two[60,65,66]. Numerous studies show the potential of using LST data to derive spatially continuous $T_a$ estimates[67–69], including LST-based estimates of $T_a$ reductions caused by urban trees. However, several examples also demonstrate the inaccuracies related to this approach, for example, in complex terrain[70], and that a better accuracy can be achieved when estimating nighttime temperatures than daytime temperatures[71]. To further increase the relevance of our results, it will be important to better understand how the spatio-temporal patterns of differences in LSTs between LULC types identified in our study translate into differences in air temperatures and other climate variables that e.g. directly influence human well-being and energy consumption in cities.

LSTs observed for different vegetation types in different regions can be largely explained by different ET levels, but LST differences do not reflect shading benefits provided by trees. Shading of trees can be particularly relevant in Mediterranean regions with high amounts of incoming solar radiation. Thus, while our results indicate where we can find larger ET-based cooling benefits in Europe, they do not show how the shading benefits vary across the continent. Thus, our results should not be interpreted as indicating the overall cooling benefits of different vegetation types in different regions. We think they are of relevance when interpreting them in combination with results produced in studies that rely e.g. on station observation and climate modelling experiments. All three approaches have their limitations in terms of spatial coverage, temporal resolution and degree of uncertainty. But looking at results from each of these approaches together can be very relevant when supporting policy making and decision-making.

In conclusion, we present an observation-based analysis of temperature differences between urban trees and urban fabric across European cities. The presented results were derived from high spatial resolution LST and LULC data from a large number of cities. Using high-resolution data at intra- and inter-city scales enabled us to demonstrate that the potential cooling benefits depend on vegetation type as well as climatic context. In general, urban trees were related to reductions in LSTs that were 2–4 times higher than the LST reduction associated with treeless urban green spaces. Both types of vegetation led to a high reduction in LSTs in Central Europe and a smaller reduction in Southern Europe. While urban trees and rural forests predominantly provided cooling in all European regions, treeless urban green spaces and rural pastures exhibited a small cooling benefit or even a warming effect in Southern European regions.

Even though vegetation within urban areas is subject to different environmental conditions and human influence than vegetation outside of cities, the cooling provided by rural vegetation and urban vegetation showed similar regional patterns in Europe. These patterns were closely related to differing ET rates across regions. In addition to regional variations, substantial seasonal variations in the cooling provided by urban trees were observed, and there was a notable influence of hot extremes. The LST reduction during hot extremes decreased in the Mediterranean and the Iberian Peninsula but increased in Scandinavia and the British Isles. In summary, our results confirm the high potential of trees to mitigate urban heat in Europe and highlight important spatio-temporal variations in their cooling effect.

## Methods

**Summary**. We use high resolution LULC data and high-resolution LST data to show temperature differences between vegetated land and urban fabric in a large number of European cities. In particular, we focus on the effect of urban trees on temperatures. To disentangle the effects of different LULC types and topography on temperature, we employ GAMs. The spatial and temporal variability of temperature differences between vegetated land and urban fabric are analysed and data on ET as well as albedo are used to test their influence on the spatio-temporal variability of temperature differences.

**Study domain**. Data on urban trees and high resolution land-cover are provided by Copernicus for cities within the administrative boundaries of the European Union and some additional countries, including Turkey (Supplementary Note 1). Instead of using all cities for which data are available, we relied on a subsample to reduce computational costs. The selection of the subsample of cities involved the following steps: First, we created a regular grid of points (50 km) over Europe and selected all cities that were lying on these grid points (234 cities). Second, in regions with a low sampling density we manually selected additional cities (47). Third, large metropolitan areas that had not previously been selected were added (12 cities). In total, the analysis included 293 cities (Supplementary Data 1). To present regional differences, we adopted a simplified categorization into European sub-areas by adding Turkey as a new region (Fig. 3b) and combining the two regions Alps and Mid-Europe into one[72].

**High-resolution urban tree, land-cover and topographic data of European cities**. The digital elevation model EU-DEM v1.0[73] was used to include elevation as a predictor and to calculate aspect information (using the function Aspect as part of the Spatial Analyst tool provided by ESRI, ArcGIS Desktop 10.5.1). The calculated aspect, which indicated the orientation of slopes (from 0° to 360°), was

reclassified into the two categories of south facing slopes (90°–270°) and north facing slopes (270°–360°, 0°–90°). Based on this information, we computed the fraction of north facing slopes for each grid cell (based on the gridded LST data). In addition to topographic attributes, we included information on LULC type based on the Copernicus urban atlas[74]. The urban atlas contains spatial polygons belonging to different LULC categories, including e.g. continuous urban fabric and green spaces (Supplementary Table 2). The polygon data were rasterized to a 10 m resolution and afterwards used to calculate the fraction of each LULC type within each grid cell. Urban tree data were available through Copernicus as an additional data set (called Street Tree Layer) to the urban atlas[74]. It includes contiguous rows or patches of urban trees covering at least 500 m$^2$ and having a minimum width of 10 m. Similar to the urban atlas data, we rasterized the street tree layer to a resolution of 10 m and afterwards calculated the fraction of tree coverage within each grid cell.

**LST data**. Two LST data sets were used to calculate temperature differences between vegetated land and the built environment of the selected European cities and their surroundings. First, Landsat LST data on 30 m resolutions were generated based on the methodology developed by Parastatidis et al.[52] and the online graphical user interface (http://rslab.gr/downloads_LandsatLST.html) provided by the authors. The methodology is based on a single channel algorithm and offers the possibility of using different emissivity sources to calculate LST values. We chose normalized difference vegetation index (NDVI)-based emissivity[75] but also tested the sensitivity of different emissivity sources for a smaller sample of cities (Supplementary Figs. 8 and 21). The estimation of NDVI-based emissivities involves three steps[52]. First, NDVI is calculated for each grid cell based on Landsat observations. Second, relying on an empirical relationship, the fraction of vegetation cover (FVC) is calculated based on NDVI values[75]. Third, the emissivity is calculate based on FVC assuming that non-vegetated surfaces have an emissivity of 0.97, vegetated surfaces have an emissivity of 0.99 and all partly vegetated surfaces are a linear combination of these two emissivities and hence lie between 0.97 and 0.99. All Landsat observations intersecting with city boundaries between 2006 and 2018 (including Landsat 5, 7 and 8) were downloaded. This resulted in at least 78 and on average 408 Landsat scenes available for each city (Supplementary Data 1). The Landsat satellite crosses every point on earth every 16 days and passes the equator approximately at 10:00 a.m. (mean local time), which results roughly in observations of European areas at 10:15. As a second data set, we included Aster (Advanced Spaceborne Thermal Emission and Reflection Radiometer) LST estimates based on the methodology developed by Gillespie, Rokugawa[76]. The data have a spatial resolution of 90 m. The Aster sensor is located on the Terra satellite and passes the equator approximately 30 min later than Landsat. Terra revolves like Aster around sun-synchronous orbit on a 16-day cycle. However, the Aster sensor is not always active and hence data coverage is in general lower than for Landsat with an average of 194 Aster scenes available per city. The data were downloaded using the earth data platform (https://search.earthdata.nasa.gov/search). To be able to compare results based on Landsat and Aster, we resampled the 30 m Landsat data to the resolution of 90 m. Both Landsat and Aster LST data were transformed into the European coordinate system ETSR89. Since there are much more observations available for Landsat, we focussed on the analysis of Landsat data and mainly used Aster data for comparison and validation (Supplementary Fig. 9).

**Albedo and ET**. Based on the MODIS albedo product MCD43A3[77] and ET product MYD16A2[78], we estimated albedo and ET of different LULC types in each city. We calculated multi-year averages (2006–2018) and aggregated the data for each month and city. As a simplified approximation of blue-sky albedo, we averaged white- and black-sky albedo[5]. This approximation is a potential source of uncertainty and bias; however, it will most likely not affect inter-city patterns of albedo differences (Supplementary Note 4 and Supplementary Figs. 14, 22 and 23). The albedo product had a resolution of 500 m and the ET product had a resolution of 1 km. To estimate the contribution of different LULC types to the observed ET and albedo values, we fitted multiple linear regression models using the fraction of each LULC type as predictor. We used the same predictors as for the models to predict LSTs (Supplementary Table 1), but the LULC fractions were calculated for the spatial resolution of Modis ET and albedo. We included all predictors in the form of linear terms. Since MODIS ET values are usually not available over urban areas, we were not able to calculate ET values for urban trees and green spaces but for forests and pastures outside of cities. Albedo and ET data are on a relatively coarse resolution and hence for small cities there is sometimes not enough data for a reliable prediction of albedo and ET values for different LULC types. In addition, in three cities the linear models predicted negative ET values, which were discarded for any further analysis. Both MODIS products (albedo/ET) have been extensively validated and show in general good agreement with ground observations, but they also show potential biases and uncertainties[79].

**Calibrating statistical models to calculate temperature differences between vegetated land and urban areas**. To calculate LST differences between different LULC categories, we use GAMs[80]. The models were fitted using the package mgcv[80] embedded in the R computing environment[81]. GAMs can be used to estimate temperatures based on a variety of predictor variables and hence can account for potential confounding factors, which has shown to be very relevant in the analysis of LULC temperature impacts[6,51]. All GAMs are calibrated including LST observations as response variable and several predictor variables (Fig. 5). These include topographic information and information on LULC type. To estimate the temperature difference between different LULC types, we use the calibrated models and make a prediction for 100% vegetation (urban trees or treeless urban green spaces) and subtract this prediction from one that estimates temperatures if 100% of a grid cell is covered by the LULC type called continuous urban fabric. The information on the location of urban tree is available as an additional layer to the LULC information. Thus, the information whether trees are located in urban parks above some form of grassland or whether they are located above sealed urban surfaces is inherent to the data. Calibrating the GAMs using the fractions of LULCs and urban trees covering each grid cell allows to estimate the signal of treeless urban green spaces even though the original LULC data (i.e. Copernicus urban atlas data) do not separate between tree-covered and treeless green spaces. The x- and y-coordinates are included as two-dimensional tensor product smooths. All other predictors are included in the form of thin plate regression splines. Including spatial coordinates as tensor product smooths reduces spatial auto-correlation and can help to reduce the potentially confounding impact of unobserved phenomena and variables[82]. Since the structure of GAMs is inherently additive, we may interpret the modelling process in a simplified way: A part of the LST signal is modelled as a function of topographic variables (e.g. elevation) and spatial location (i.e. x–y-coordinates) and the remaining signal is expressed as a function of the land-cover at a specific location. However, it should be noted that, while the effect of the different land-covers is modelled based on smooth functions (i.e. nonlinear functions), we do not model the effect as a spatial interaction term. This means we are interested in the average effect of e.g. urban trees over the whole city and not in specific patterns within each city. This is justified by the scale of our analysis looking at inter-city differences, but of course intra-city differences can be equally important. This model set-up was complemented by sensitivity experiments to

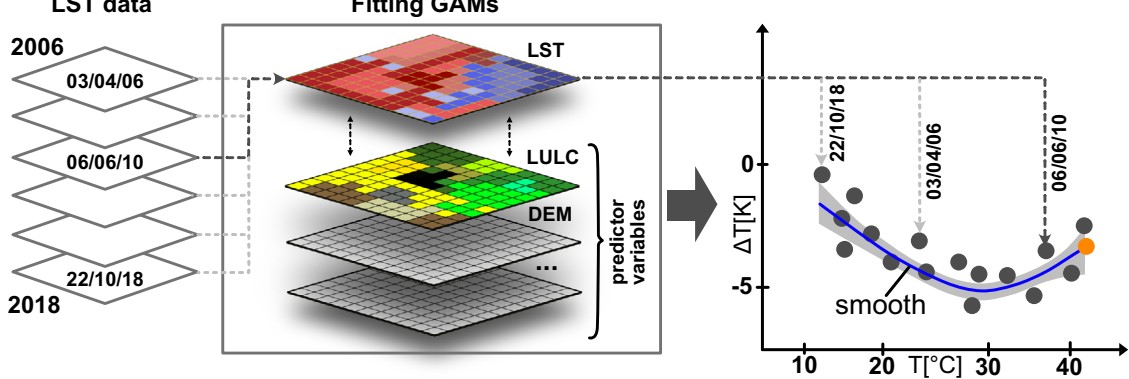

**Fig. 5 Schematic of modelling process showing conceptually how the LST observation are analysed in each city.** LST observations between 2006 and 2018 are used as response variables when fitting GAMs including several predictor variables (e.g. DEM—Digital Elevation Model). The resulting models are used to make predictions on the temperature difference between vegetated land and continuous urban fabric. A smooth function is fitted to approximate these differences based on background temperature and the value indicated by the smooth for the highest observed temperature (orange dot) is used for a comparison with other cities. The grey area indicates the uncertainty in the form of a confidence interval.

show how the model set-up influences the results (Supplementary Note 2 and Supplementary Figs. 11 and 12). For some model set-ups, the effect of trees in certain cities could not be estimated in a numerically stable manner or the results were not significant. Such cities were removed from the sensitivity analysis (38 cities). All fitted models showed a decent coefficient of determination ($R^2$), which averages to 0.64 considering all cities. The $R^2$ in Turkey was lower in comparison to other European regions (Supplementary Fig. 10).

**Estimating LST differences between vegetation and urban fabric for varying conditions**. We fit a GAM for each LST observation available to be able to distinguish the potential cooling effect of urban vegetation for varying conditions (e.g. varying background temperatures). Since there is a separate GAM for each observation, not only the effect of vegetation on temperature but also the effect of all variables is estimated separately for each observation. We use E-OBS (v20.0e) temperature data[83] as an indicator for the background temperature. The gridded data set of air temperature is based on station data. It is available for all European cities except for some cities in Turkey. In cities, in which E-OBS data are not available for the whole period of 2006–2018 we calculate the spatial average of each satellite observation as an indicator for the background temperature. We plot temperature differences (e.g. between urban trees and urban fabric) against the background temperature estimated based on E-OBS and use LOESS (locally estimated scatterplot smoothing) to estimate a smooth (loess) curve through all data points (Fig. 5 and Supplementary Fig. 18). Instead of using least-squares for fitting the smooth, we rely on a more robust fit based on a re-descending M-estimator as implemented in the loess function of the stats package in R[84]. The last point of the loess curve is considered as the temperature difference between vegetated urban land and urban fabric for the hottest and hence most extreme observation available.

To analyse the spatial variation of the LST differences between vegetated land and urban fabric, we calculated smooth spatial trends of the LST differences. The smoothing along geographic coordinates was carried out using GAMs and can be interpreted as an interpolation of the LST differences calculated for each city. The LST differences available for all cities within a specific European country were also summarized by their mean and standard error of the mean. For selected cities, LST differences between vegetation and urban fabric are shown for different seasons and as a comparison of summertime average and hot extreme conditions (details on how these cities were selected can be found in Supplementary Note 6).

## Data availability

The data on LST differences generated in this study have been deposited on zenodo and are publicly available at https://doi.org/10.5281/zenodo.5526674. These data include estimates of the LST differences between urban fabric, urban trees and urban green spaces for each city and the LST differences between urban fabric, rural forests and rural pastures. In addition, estimates of the evapotranspiration of forests and pastures of each city and albedo estimates of urban fabric and forests are provided. All additional data are available from the following sources: Landsat LST can be retrieved from http://rslab.gr/downloads_LandsatLST.html and Aster LST from https://search.earthdata.nasa.gov/search. EU-DEM v.1.0 can be downloaded from https://land.copernicus.eu/imagery-in-situ/eu-dem/eu-dem-v1-0-and-derived-products/eu-dem-v1.0, the Copernicus Urban Atlas and Street tree data from https://land.copernicus.eu/local/urban-atlas, E-OBS gridded data from https://www.ecad.eu/download/ensembles/download.php and MODIS albedo and ET data from https://search.earthdata.nasa.gov/search.

## Code availability

Code providing details on how LST differences were calculated is available at https://zenodo.org/record/5526734#.YU3epH2xWUk.

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

## Acknowledgements

We acknowledge funding from the Swiss National Science Foundation (SNSF) and the Swiss Federal Office for the Environment (FOEN) through the CLIMPULSE project (http://p3.snf.ch/Project-172715; grant no. 200021_172715). In addition, we would like to thank Diego Schnyder and especially Jan Mathias for their help in collecting some of the data.

## Author contributions

J.S. and E.L.D. conceptualized the study. E.L.D. acquired the funding and supervised the project. J.S. conducted the analysis and produced the figures with help from C.B. and G.M. E.L.D, S.I.S., G.M. and R.M. critically assessed the results, helped with the interpretation and encouraged additional analysis. J.S. wrote the manuscript with the help of all authors.

## Competing interests

The authors declare no competing interests.
