## [Peer Review File · Nature Communications]

REVIEWER COMMENTS

Reviewer #1 (Remarks to the Author):

Summary:

The present study, "The Role of Urban Trees in Mitigating Heat in European Cities" examines the influence of urban trees on satellite-derived land surface temperature (LST) across 293 European cities. They find that the efficiency with which trees reduce LST depends on the region and background conditions, for instance, trees having a different cooling influence during heat waves. Moreover, trees have a stronger effect on LST than treeless green space. While the results themselves are not particularly surprising, I can see how this data can have policy implications. However, to do so, several improvements need to be made with reference to the methodological and conceptual framework to make the results applicable to urban decision making.

Major comments:

1. The idea of heating and cooling is misleading when examined using satellite-derived LST. LST is not the same as near-surface air temperature, which can be several degrees cooler than LST during daytime. LST is closer to air temperature at night, which this study does not focus on. More importantly, the coupling between LST and air temperature is a function of, among other things, vegetation cover [Mildrexler, et al. 2011]. In general, one would expect the coupling to be stronger i.e. air temperature to be closer to surface temperature for vegetated surfaces during the day, while LST would be much higher than air temperature over barren land, including built-up structures in cities (though satellite-derivations of these structures also include rooftop and wall temperatures). Therefore, any cooling signal on LST due to urban trees is likely an overestimation of actual cooling of near-surface temperature, as has also been seen in previous studies comparing surface urban heat island (SUHI) intensity with air temperature or canopy urban heat island (CUHI) intensity [Hu et al. 2014; Zhang et al. 2019]. As an example, Novick and Katul [2020] found that the cooling effect of forests on air temperature was half as strong as the cooling effect on surface temperature for a temperate site. This does not suggest that the results are useless since cooling surface temperature is also important. However, it is less important when framing the cooling in the context of public health and energy consumption. Ideally, I would like to see the authors include air temperature (and even humidity) in this analysis, though I know that urban weather measurements are hard to get and are never spatially continuous. Another option is to discuss these limitations in detail and how they relate to the main results and implications of the study. Such discussions are quite common in the SUHI literature due to the same limitations of using LST to isolate the urban impact on local temperatures [Chakraborty et al. 2020]. I would also suggest that authors add disclaimers when mentioning the numbers for the cooling effect of trees, which are noted prominently throughout the manuscript including the abstract, since they are likely biased high, and can thus be misleading for policy makers. Finally, maybe changing the title to 'The Role of Urban Trees in Mitigating Surface Temperature in European Cities' might be more appropriate.
2. Clouds are another issue that the study mentions once in the limitations, but requires further clarification and, potentially, analysis. Landsat images, since they are available every 16 days are quite prone to cloud cover. I am concerned how this leads to sampling biases among the cities. Based on the data from 2006 to 2018, is the seasonality of clouds, and thus, missing data different for the 293 cases? Specifically, does this difference in seasonality confound any actual regional differences? I think the authors need to give an estimate of missing data in the Landsat images by month for each city and check if they are confusing temporal and spatial effects. I can see large differences in the number of Landsat images for cities in Table S1, though cannot tell from those if there is a seasonal bias. Additionally, it is important to know whether the missing data fraction varies between the continuous urban fabric and the tree-covered pixels. For MODIS, the missing data percentage tends to be higher over urban areas due to uncertainties, not all of which relate to clouds.
3. The authors mention that they use black-sky albedo as a simplification for blue-sky albedo. However, the diffuse fraction can be quite high for some European cities. It might be useful to check whether blue-sky albedo and black-sky albedo are relatively close across all land cover types and all cities considered.
4. The authors consider the mean LST of the continuous urban fabric and the tree-covered pixels. However, that does not tell us the location of urban forests within each city, which can affect the LST differences [Zhou et al. 2011]. There are no large-scale studies on the configuration of urban forests and their impact on the urban LST and I think analyzing that would make this study more impactful.

Minor comments:

1. Line 29: Many studies on satellite-derived UHI, in fact, the bulk of the studies, consider the linear and sometimes non-linear relationships between LST and some proxy for vegetation cover, usually the Normalized Difference Vegetation Index (NDVI). Vegetation cover, particularly its cooling effect, does lie on a continuum. By considering tree and treeless green space, the study here focuses on the two points of this distribution. If the reason for doing so is to quantify the magnitude of the effect for these binary cases, which, in my opinion, is less useful for studying the climatic effects of real cities than the continuous distribution used in other studies, based on my first major comment, the authors would have to add sufficient disclaimers to those numbers. A possible way to reframe this is in the context of policy, which the authors mention in the final line of the abstract but do not discuss in detail, is to state that while NDVI is useful for studying the climatic effects of urbanization from the evaporative cooling perspective, it is hard to implement NDVI-based policy solutions. Rather, urban forests versus urban parks are easier to implement, which is why the authors focus on the two points of the continuous distribution.
2. Line 33: There are studies that look at the temperature of different LU/LC within urban areas [Silva et al. 2018] I would reframe this to the focus on the two primary types of urban vegetation.
3. Line 38: It is true that tree species composition varies across cities. However, the authors never address this in the actual study. I would avoid talking about it here or add a discussion on this based on the kinds of trees one expects in the different regions considered.
4. Line 130: This is somewhat of an exaggeration. There are multi-city observational studies looking at the effects of urban trees on urban LST [Kroeger et al. 2018]. I would avoid this line.
5. Table S2: Landsat is NASA+USGS mission, while Aster is a NASA+METI mission.

References:

- Mildrexler, D. J., Zhao, M., & Running, S. W. (2011). A global comparison between station air temperatures and MODIS land surface temperatures reveals the cooling role of forests. *Journal of Geophysical Research: Biogeosciences*, 116(G3).
- Novick, K. A., & Katul, G. G. (2020). The Duality of Reforestation Impacts on Surface and Air Temperature. *Journal of Geophysical Research: Biogeosciences*, 125(4), e2019JG005543.
- Hu, Y., Hou, M., Jia, G., Zhao, C., Zhen, X., & Xu, Y. (2019). Comparison of surface and canopy urban heat islands within megacities of eastern China. *ISPRS Journal of Photogrammetry and Remote Sensing*, 156, 160-168.
- Zhang, P., Bounoua, L., Imhoff, M. L., Wolfe, R. E., & Thome, K. (2014). Comparison of MODIS land surface temperature and air temperature over the continental USA meteorological stations. *Canadian Journal of Remote Sensing*, 40(2), 110-122.
- Chakraborty, T., Hsu, A., Manya, D., & Sheriff, G. (2020). A spatially explicit surface urban heat island database for the United States: Characterization, uncertainties, and possible applications. *ISPRS Journal of Photogrammetry and Remote Sensing*, 168, 74-88.
- Zhou, W., Huang, G., & Cadenasso, M. L. (2011). Does spatial configuration matter? Understanding the effects of land cover pattern on land surface temperature in urban landscapes. *Landscape and urban planning*, 102(1), 54-63.
- Silva, J. S., da Silva, R. M., & Santos, C. A. G. (2018). Spatiotemporal impact of land use/land cover changes on urban heat islands: A case study of Paço do Lumiar, Brazil. *Building and Environment*, 136, 279-292.
- Kroeger, T., McDonald, R. I., Boucher, T., Zhang, P., & Wang, L. (2018). Where the people are: Current trends and future potential targeted investments in urban trees for PM10 and temperature mitigation in 27 US cities. *Landscape and Urban Planning*, 177, 227-240.

Reviewer #2 (Remarks to the Author):

Review for Manuscript#: NCOMMS-20-42315

This is a neat piece of writing. The authors present temperature differences between urban trees, urban non-tree green spaces, and urban fabrics in summer and extreme heat. So, this is a new perspective on integrated urban climate studies, as previous studies have typically compared urban-rural temperature differences. The analysis is mainly based on high-resolution land use/land cover data, Landsat (30m) and ASTER (90m) LST data. The technique for constructing suburban details from satellite observations is impressive. In the Methods section, I note that this is based on a Generalized Additive Model, which appears to be a statistical approach. It is important to explain some of the spatial and temporal assumptions for this method. I wondered about the uncertainties regarding these assumptions, especially with respect to LST during the summer day and during these extremely hot days.

The description of the LST phenomenon is fairly straightforward. However, explanations of the mechanisms behind the phenomena lack evidence or require a deeper understanding (e.g., in the section on biophysical processes in the discussion, these explanations are possible, but not sufficient). The results may have implications for policymakers, but until the theory of mechanisms is understood, their usefulness for mitigating future climate change may be rather uncertain.

In addition, the interpretation of biophysical processes is based on spatial substitution (e.g., L84-88, L192-200). That is, LST phenomena are observed and simulated at the suburban scale, whereas biophysical processes are explained by replacing urban trees with rural forests and urban nontree green spaces with rural pastures. It seems that one should be concerning of this, as their correlation (R^2) in Figs.S1a and b is only about 0.6. Such a hypothesis should be aware of the uncertainty. Also, I wonder if the authors can reconstruct the urban details of albedo, roughness, ET, and other biophysical parameters of urban trees, nontree green spaces, and urban fabrics?

Minor comments:

L61: Why does Gaziantep in Turkey show a higher LST for urban trees than for urban fabrics? Perhaps a note could be added to the discussion.

L58, 70, 72: Do "hot temperature extremes", "hot days", and "heat extremes" refer to these orange dots in Fig2b? It would be better to use a consistent statement.

L65-75: Interesting. Is it because of the stomatal control of trees? Linking this to L61, I wonder what the leaf area index (or soil exposure and canopy openness) is for "urban trees" in Gaziantep. Is it possible that Gaziantep's "urban trees" are evaporating much more than they are transpiring?

L76: Should the temperatures in Fig2c mostly be negative in sign, according to the definition in your figure caption.

For the Results section: It would be better to specify what LST is presented at the section beginning. For example, 24-hr average, daytime average, nighttime average.

L86-88: In the main text, it is ET over pastures; in the caption of Fig 4b, it is ET over forests. Which is true? In addition, why R^2 in Fig 4b is higher than R^2 in Fig 4a?

L155: How do you determine the emissivity from remote sensing data for "urban trees", "green space", and "urban fabric"?

L166-170: Spatially averaging or temporally averaging of LST? Also, these two sentences are difficult to understand precisely. Could the author have made it clearer?

Reviewer #3 (Remarks to the Author):

Review of "The Role of Urban Trees in Mitigating Heat in European Cities" by Schwaab et al. (NCOMMS-20-42315).

The authors investigate the cooling effect of urban trees in 293 European cities combining high resolution Land Surface Temperature (LST) observations with land use/land cover (LULC) data and a detailed Street Tree layer from the Copernicus Urban Atlas. They find that, during heat extremes, urban trees are 7-10 K cooler than urban surfaces in central Europe but only 0-4 K cooler in southern regions (0-4 K). They also show that LST differences between urban trees and treeless green spaces can reach 6 K. This study provides novel insights on the cooling effect of urban greening during average versus extreme temperatures and, especially, on the differences between tree and treeless urban green spaces. However, I have some concerns on the methodology/quality of the paper (see comments below) and, in my opinion, the manuscript needs significant improvements before it can be considered for publication in Nature Communications.

General Comments

- First of all, the manuscript structure/content should be improved. Specifically:
 - o The introduction is too succinct and does not provide a sufficient overview of the existing literature on the effects of vegetation on urban climate (see, for example, Wang et al. (2018, 2019), Rahman et al. (2020), Winbourne et al. (2020) and references therein). I think the manuscript would benefit from a brief (but detailed) discussion on key processes and main results in the literature (e.g. shading/evapotranspiration, observed air/surface temperature changes).
 - o The article has no conclusions – it ends with a "limitations" section and the reader is left wondering what the take-home message is.
 - o In general, the language should be improved (there are several repetitions, see specific comments below).

- One of the most interesting results here is the quantification of tree cooling versus the effect of other green spaces. However, the manuscript falls short in providing some key information thus diminishing the overall quality of the results/discussion. Specifically:
 - o The authors should clarify in the main text how the different land cover types have been defined. Do "street trees" include urban parks, urban forests, etc? What is considered "green spaces"? Are street trees included in the "green spaces" or not? What about forests and pastures? For example, do pastures include rural crops, grass, etc? I think the manuscript would benefit from some clarifications (e.g. distinguishing between trees, grass, shrubs or considering LAI). The authors often refer to "different vegetation types" (e.g. line 2018) but these are not clearly defined.
 - o The authors should clarify (i.e. using the same symbols and terminology) which variables are related to "trees" and which ones are related to "other green spaces". For example, in Fig 1 it would be useful to replace ΔT with ΔT_{trees} and ΔT_{green} or use the same subscripts employed in Figure S1-S2. It should also be clarified in the main text how temperature differences are computed. In other words, do they represent the difference between the average temperature of all the "tree/green" pixels and the average of all the "continuous urban fabric" pixels (i.e. one value per city) or is ΔT a spatially distributed variable (as, for example, in the Global SUHI Explorer, <https://yceo.users.earthengine.app/view/uhimap>)? This is partly addressed in the Methods section but it is not fully clear. For example, does the "spatial variation of the LST differences between vegetated urban land and urban fabric" in Line 360 refer to intra-urban or regional (city-to-city) variations? Some clarifications are needed to help the reader understand the methodology upfront (i.e. before the results section).
 - o I think there are some missed opportunities here. For example, an overview of the percentage of tree cover/green spaces in the analysed cities would be quite interesting and straightforward to show. In general, results are shown for different cities/regions/countries, which is interesting but limits the generalizability of

the conclusions. What about possible trends with urban or background climate characteristics (see specific comments below)?

o A clear discussion on the key biophysical mechanisms (e.g. ET, water stress, albedo, roughness) causing the observed changes in LST is somewhat missing. This is partly addressed at the end of the discussion section but the arguments are rather qualitative and it is unclear whether they refer to city-scale or street canyon conditions (e.g. at the canyon level urban trees can increase or decrease roughness depending on building height, while at the city level the impact of trees on urban roughness is probably negligible).

- My major concern is about the use of DEM/Aspect data from Copernicus. Are they representative for cities? I am not familiar with the details of this dataset but I doubt that it includes information on urban morphology (which is quite complex at the resolution considered here). Global scale datasets of building heights and 3D urban structures are now available (e.g. Li et al. 2020) and, in my opinion, should have been employed here. In other words, topography can play a role in a city like Zurich but, in general, the intra-urban variability of LST is largely controlled by the 3D structure of the urban fabric and its impact on the surface energy fluxes. These are key considerations that seem to be overlooked. It is also not fully clear to me how ET and albedo data have been used (see specific comments below).

Specific comments

Line 16: "different pattern", explain better

Lines 25-28: repetition "Based on". Also, I guess this part should be on "urban vegetation", thus justifying the following sentence on the lack of studies distinguishing between different vegetation types (e.g. trees vs no trees).

Line 29, 32, etc: language can be improved.

Lines 39-40: "urban areas" is repeated 4 times here. Please rephrase to improve readability.

Lines 71-73: "is clearly lower ... In contrast, the difference (...) is low", this sentence is unclear. Please clarify.

Line 85-88: what about the correlation between LST differences of urban trees and the ET of rural pasture and the LST of urban green spaces and the ET of rural forests? They might be correlated as well (see other comments below). Actually, if possible, it would be interesting to compare the characteristics of urban and rural vegetation (e.g. considering LST or albedo given that ET is not available for urban areas here).

Line 134: typo, "that that"

Lines 143-145: this is quite obvious – the cooling effect of urban green spaces (even if treeless) is clearly different from that of urban fabric.

Lines 146-149: this is interesting – is it shown somewhere?

Lines 168-170: this is also very interesting – some references/information on this debate would be useful (i.e. is there any evidence supporting one or the other viewpoint?)

Lines 174-175: "water scarcity", what about the projections for the regions where trees seem to be more important? A discussion or, at least, a reference would help.

Line 182: is it associated with ET of rural forests or ET of urban trees? What is generating urban cooling is the ET of urban trees which should be comparable with that of rural forests. However, UHI are generally defined as urban-rural differences, so that changes in rural ET can also modify their magnitude. Please clarify.

Line 184-187: this is not a hypothesis, impermeable urban surfaces have negligible ET. This and the following discussion are somewhat obvious (or just unclear to me).

Lines 204-209: the discussion on roughness is unclear. For example, does "roughness of forests" (line 205) refer to urban or rural forests? In cities, given the limited spatial extent of green spaces, roughness is largely controlled by the urban fabric. Also, there are many studies on the subject that should be considered (e.g. Zhao et al. 2014, Li et al. 2019).

Lines 235-236: I agree – why this was not included in the analysis? See previous comment on the availability of building height datasets.

Line 254: what is the reason for selecting cities by creating a regular grid?

Lines 265-275: do topography/aspect calculations refer to natural or urban surfaces? How is this information used? I guess the complex 3D structure of the urban fabric is not accounted for so I am not sure how topography and aspect can be estimated for urban areas at a 10m resolution (see general comments above).

Lines 312-313: Is the "contribution of different LULC types to the observed ET and albedo values" illustrated somewhere?

Lines 316-317: why did the model provide negative values? How reliable are the results for the other cities? In general, it should be clarified how the regression models were produced.

Line 337: define R2

Line 346: how is "background temperature" defined? Is it the "spatial mean LST of each satellite observation" (line 355)? I guess this might be affected by the fraction of urbanized area of the Landsat image

Line 582: typo, "large"

Figure 1: The figure is quite catchy but not fully explicative to me. For example, given that the stacked bars are illustrated without the y-axis, the magnitude of the error bars is not directly quantifiable. Also, the map is very interesting but little visible. And what is B & H? Please revise and/or clarify.

Figure 2: how were the 22 cities selected? What is the ordering criteria for the x-axis in panel b? Why not showing an urban or climatic variable there (e.g. urban area, background temperature or similar)? In panel c, a different colorbar would help (the transition from negative to positive values is not clearly distinguishable). And what about the cities in panel d? Are they exemplary cities? It might be useful to highlight them on the map (e.g. in panel C). Also, please write Athens (and not Athina).

Figure 4: I guess there is a typo in the legend, line 124 – is ET in panel b for forests or pastures (see main text)? Maybe, clarify this also on the axis (e.g. ET_forest, ET_pasture and DeltaT_trees, DeltaT_green).

Figure 5: this is very nice. Does the DeltaT-T relation always have that shape? Is the relation based on data or is it a conceptual representation? If it is a conceptual diagram, it would be useful to see some examples of "real data" (e.g. as additional figures in the SI).

Figure S1: what about the correlation between green spaces and forest as well as urban trees and pasture? I don't think that they will be uncorrelated – if that's the case, why should forest be a predictor for urban trees and pasture for other green spaces?

Figure S2: what is the x-axis? It would be nice to see if there is a trend with some specific (e.g. climatic)

variables

Figure S3: it would be useful to see an actual comparison here (i.e. STL vs Lidar).

Figure S6: write Athens and Bucarest (no Athina and Bucuresti). This should be checked throughout the manuscript.

References

Li, D., Liao, W., Rigden, A. J., Liu, X., Wang, D., Malyshev, S., & Shevliakova, E. (2019). Urban heat island: Aerodynamics or imperviousness?. *Science Advances*, 5(4), eaau4299.

Li, M., Koks, E., Taubenböck, H., & van Vliet, J. (2020). Continental-scale mapping and analysis of 3D building structure. *Remote Sensing of Environment*, 245, 111859.

Rahman, M. A., Stratopoulos, L. M., Moser-Reischl, A., Zölch, T., Häberle, K. H., Rötzer, T., ... & Pauleit, S. (2020). Traits of trees for cooling urban heat islands: A meta-analysis. *Building and Environment*, 170, 106606.

Wang, C., Wang, Z. H., & Yang, J. (2018). Cooling effect of urban trees on the built environment of contiguous United States. *Earth's Future*, 6(8), 1066-1081.

Wang, C., Wang, Z. H., Wang, C., & Myint, S. W. (2019). Environmental cooling provided by urban trees under extreme heat and cold waves in US cities. *Remote sensing of environment*, 227, 28-43.

Winbourne, J. B., Jones, T. S., Garvey, S. M., Harrison, J. L., Wang, L., Li, D., ... & Hutyra, L. R. (2020). Tree Transpiration and Urban Temperatures: Current Understanding, Implications, and Future Research Directions. *BioScience*, 70(7), 576-588.

Zhao, L., Lee, X., Smith, R. B., & Oleson, K. (2014). Strong contributions of local background climate to urban heat islands. *Nature*, 511(7508), 216-219.

We thank all reviewers for their time and effort and their very helpful and constructive comments. In the following, we list the reviewers' comments in black and our reply in blue. Please note that we slightly changed the numbering of the comments to make it easier to refer to specific comments (i.e. to make cross references) when answering each of them. Comments of reviewer 1, 2 and 3 are referred to as comments 1.1, 1.2 ... 2.1,2.2... 3.1,3.2 etc. In addition, it should be noted that we reference all figures included in this response according to the numbering of the figures in the revised manuscript.

Reviewer #1 (Remarks to the Author):

Summary:

The present study, "The Role of Urban Trees in Mitigating Heat in European Cities" examines the influence of urban trees on satellite-derived land surface temperature (LST) across 293 European cities. They find that the efficiency with which trees reduce LST depends on the region and background conditions, for instance, trees having a different cooling influence during heat waves. Moreover, trees have a stronger effect on LST than treeless green space. While the results themselves are not particularly surprising, I can see how this data can have policy implications. However, to do so, several improvements need to be made with reference to the methodological and conceptual framework to make the results applicable to urban decision making.

Major comments:

1.1) The idea of heating and cooling is misleading when examined using satellite-derived LST. LST is not the same as near-surface air temperature, which can be several degrees cooler than LST during daytime. LST is closer to air temperature at night, which this study does not focus on. More importantly, the coupling between LST and air temperature is a function of, among other things, vegetation cover [Mildrexler, et al. 2011]. In general, one would expect the coupling to be stronger i.e. air temperature to be closer to surface temperature for vegetated surfaces during the day, while LST would be much higher than air temperature over barren land, including built-up structures in cities (though satellite-derivations of these structures also include rooftop and wall temperatures). Therefore, any cooling signal on LST due to urban trees is likely an overestimation of actual cooling of near-surface temperature, as has also been seen in previous studies comparing surface urban heat island (SUHI) intensity with air temperature or canopy urban heat island (CUHI) intensity [Hu et al. 2019; Zhang et al. 2014]. As an example, Novick and Katul [2020] found that the cooling effect of forests on air temperature was half as strong as the cooling effect on surface temperature for a temperate site. This does not suggest that the results are useless since cooling surface temperature is also important. However, it is less important when framing the cooling in the context of public health and energy consumption. Ideally, I would like to see the authors include air temperature (and even humidity) in this analysis, though I know that urban weather measurements are hard to get and are never spatially continuous. Another option is to discuss these limitations in detail and how they relate to the main results and implications of the study. Such discussions are quite common in the SUHI literature due to the same limitations of using LST to isolate the urban impact on local temperatures [Chakraborty et al. 2020]. I would also suggest that authors add disclaimers when mentioning the numbers for the cooling effect of trees, which are noted prominently throughout the manuscript including the abstract, since they are likely biased high, and can thus be

misleading for policy makers. Finally, maybe changing the title to ‘The Role of Urban Trees in Mitigating Surface Temperature in European Cities’ might be more appropriate.

>>>> We agree that it is important to emphasize better that land-surface temperature is different from near-surface air temperature. To clarify this point more prominently, we followed your suggestion and changed the title to: “The Role of Urban Trees in Reducing Land Surface Temperatures in European Cities”. In addition, we added a more extensive discussion of this limitation in particular stressing that air temperature differences between different land-cover types are expected to be in general smaller than LST differences. We also clarify that satellite observations are by design also influenced by rooftop- and wall-temperatures, which could potentially lead to higher temperature differences than would otherwise be observed if the analysis had been strictly limited to street canyon conditions. It is, however, important to mention that this is equally true for all cities and is therefore unlikely to strongly affect the observed spatio-temporal patterns at European scale which is the primary focus of our study.

We also want to mention that our choice of not including analysis of 2m air temperature is linked to current data limitation. Indeed, in-situ observations of 2m air temperature and humidity are available for whole Europe through at least two databases (<https://www.ncdc.noaa.gov/isd> and <https://www.ecad.eu/>). However, we discovered that the location of a majority of the station observations is not provided with sufficient precision and can differ by up to 1000 m from the true location. These deviations make it extremely difficult to extract robust correlations between land-cover/vegetation and temperature. In addition, the amount of stations in urban areas is still quite low. While we are in contact with both data providers and hope that the data will be improved in the future, it is currently too premature to include an analysis of air temperature.

Added to discussion:

“Our analysis is focused on LST, which is less directly related to the adverse impacts of urban heat than air temperatures. The relationship between LST derived from satellite observations and air temperature (T_a) is complex (Martilli et al., 2020, Manoli et al., 2020b, Chakraborty et al., 2020). Under cloud-free and low wind speed conditions in summer, daily maximum LSTs are usually several degrees higher than air temperatures (Zhang et al., 2014, Good, 2016). This is particularly the case over agricultural and barren land but less so over forested land (Mildrexler et al., 2011). Accordingly, it has been shown that differences in T_a between forests and grassland are smaller than differences in LST between these two land-covers (Novick and Katul, 2020). Likewise, the surface urban heat island (SUHI), being based on LST estimates, is often higher than the canopy urban heat island (CUHI), which is based on T_a air temperature estimates (Zhang et al., 2014, Hu et al., 2019). While there are clear systematic differences between LST and T_a , there are also clear correlations between the two (Zhang et al., 2014, Hooker et al., 2018, Serra et al., 2020). Numerous studies show the potential of using LST data to derive spatially continuous T_a estimates (Benali et al., 2012, Kloog et al., 2012, Alonso and Renard, 2020), including LST-based estimates of T_a reductions caused by urban trees. However, several examples also demonstrate the inaccuracies related to this approach, for example, in complex terrain (Mutiibwa et al., 2015) and that a better accuracy can be achieved when estimating nighttime temperatures than daytime temperatures (Ho et al., 2016). To further increase the relevance of our results it will be important to better understand how the spatio-temporal patterns of differences in LSTs between LULC types

identified in our study translate into differences in air temperatures and other climate variables that influence human well-being and socio-economic factors.”

1.2) Clouds are another issue that the study mentions once in the limitations, but requires further clarification and, potentially, analysis. Landsat images, since they are available every 16 days are quite prone to cloud cover. I am concerned how this leads to sampling biases among the cities. Based on the data from 2006 to 2018, is the seasonality of clouds, and thus, missing data different for the 293 cases? Specifically, does this difference in seasonality confound any actual regional differences? I think the authors need to give an estimate of missing data in the Landsat images by month for each city and check if they are confusing temporal and spatial effects. I can see large differences in the number of Landsat images for cities in Table S1, though cannot tell from those if there is a seasonal bias. Additionally, it is important to know whether the missing data fraction varies between the continuous urban fabric and the tree-covered pixels. For MODIS, the missing data percentage tends to be higher over urban areas due to uncertainties, not all of which relate to clouds.

>>>> Thank you for highlighting this important point. We now tried to better understand potential sampling biases and included an extensive discussion together with several new/modified figures in the supplementary material (e.g., Figure S 1, Figure S 13) and a modified figure in the main part including uncertainty estimates (Figure 2). In addition, we emphasize in the manuscript that our results should be strictly interpreted as for cloud-free conditions.

We added the following part to manuscript (Supplementary Material):

“Due to seasonal variations in cloud-cover, there are more Landsat observations available in summer than in winter (Figure S 13 a, b). The number of observations in winter is particularly low for cities in Scandinavia and comparably high for cities of Turkey, the Iberian Peninsula and the Mediterranean. This increases the uncertainty in winter, but is of minor importance since we are focusing on summertime temperatures. To better understand potential sampling biases in summer, we used E-OBS (v20.0e) temperature data (Cornes et al., 2018), which is a gridded dataset of air temperature based on station data and available for all conditions. By separating all E-OBS observations for each city into quantiles, we separated dates when low temperatures were observed from days when hot temperatures were observed. These dates were matched with the dates, when Landsat observations were available, to find out whether Landsat data were missing for certain quantiles. The regionally summarized results show that for cities in all regions observations are available for low and high quantiles in summer (Figure 13 c). However, in some regions (e.g. France and British Isles) there are much less observations available for low quantiles than in other regions (e.g. Turkey). We also tested whether LST data was more often available for certain LULC types than for others, but did not find substantial differences (Figure 13 d).

Cloudiness clearly leads to data gaps and highlights that our results should be strictly interpreted as being only valid for cloud-free conditions. Since observations for high temperatures are consistently and frequently available for all regions, the smooth functions that are used to estimate temperature differences between different LULC types, are relatively robust and it is unlikely that we strongly confound temporal and spatial effects. Missing data for colder and presumably cloudy days in summer will, however, have an impact on average summertime temperature differences between different land-covers. Since the differences in the amount of observations during hotter and colder days in summer are relatively small in Scandinavia, Turkey, Iberian P./Mediterranean, it can be assumed that

our observations would allow relatively robust conclusions about the full/true climatology in these regions. In regions such as the British Isles, France, Eastern Europe and Mid-Europe, including temperature differences during cloudy days (which are presumably small) will have an important impact on summertime mean temperature differences between different LULC types and hence may cause some bias.”

Figure S 13: Analysis of temporal and spatial sampling biases. a) Number of observations summarized for different European regions and four seasons (DJF, MAM, JJA and SON). b) Number of observations available in each month. c) Number of days with Landsat observations available for each E-OBS quantile in summer. d) Fraction of each land-cover type that is observed in different regions.

Besides testing these potential sampling biases, we also made a major methodological change (in response to this comment (1.2) and in response to reviewer comment 3.29). Instead of using the spatial average of LST to define background temperatures, we now use the gridded air temperature data E-OBS (v20.0e) (Cornes et al., 2018) to define background temperatures. This is possible for almost all cities, but particularly in Turkey the E-OBS data is not consistently available for all cities. For cities for which

the E-OBS data is not available, we fall back to the previous approach of spatially averaging the LST observations.

In addition, we applied a more robust smoothing and we corrected a mistake that we have noted, which was a procedure that would check for outliers but accidentally removed plausible values in cities in central Europe that showed large temperature differences between urban trees and urban fabric. All these changes together did have some major effect on the results, in particular on the pattern of temperature differences between extreme and average conditions (i.e. on former Figure 2, now Figure S 1). We updated all figures in the manuscript accordingly and also respective parts of the results and discussion.

1.3) The authors mention that they use black-sky albedo as a simplification for blue-sky albedo. However, the diffuse fraction can be quite high for some European cities. It might be useful to check whether blue-sky albedo and black-sky albedo are relatively close across all land cover types and all cities considered.

>>>> We used an average of white- and black-sky albedo to partly account for the fact that the diffuse fraction of incoming radiation could be indeed important. Even though this approach has been frequently used (e.g. Li et al., 2015, Duveiller et al., 2018), we agree that it is a simplification. Since the blue-sky albedo (Lewis and Barnsley, 1994, Wang et al., 2015) can be approximated as a linear combination of black- and white-sky albedo, the value of the blue-sky albedo will usually lie between the min/max values of black- and white-sky. Hence to show potential uncertainties we calculated albedo values only based on white-sky albedo and only based on black-sky albedo. While there are differences, the spatial pattern of the albedo differences seems to be very robust (Figure S 14). We modified the methods section and included the following into the supplementary material:

“Albedos of continuous urban fabric in different regions were calculated separately for white- and black-sky albedo (WSA/BSA). WSA values are generally higher, however, we find that the regional patterns are the same for WSA and BSA (Figure S 14). Since BSA and WSA show very similar regional trends, it seems rather unlikely that blue-sky albedo values will strongly deviate from these trends. However, if diffuse radiation over cities in certain European regions is much larger (e.g., due to air pollution) than in other regions, there may be slight changes in these trends, compared to the ones we see when weighing WSA and BSA equally.”

Figure S 14: Black Sky Albedo (BSA) and White Sky Albedo (WSA) of continuous urban fabric in different European regions.

1.4) The authors consider the mean LST of the continuous urban fabric and the tree-covered pixels. However, that does not tell us the location of urban forests within each city, which can affect the LST differences [Zhou et al. 2011]. There are no large-scale studies on the configuration of urban forests and their impact on the urban LST and I think analyzing that would make this study more impactful.

>>> We thank the reviewer for this suggestion. However, we think that adding an analysis of the spatial configuration of the urban forests (together with a large body of additional methodology) would add too much information to the manuscript and reduce its readability. Nevertheless, we acknowledge that such additional analysis would be an interesting follow up of this work and added the following sentences to the discussion.

“... . The effects of different LULC configurations may also be included directly, for example, by using landscape metrics (Schwarz and Manceur, 2015, Debbage and Shepherd, 2015, Zhou et al., 2011). In addition, the analysis of Local Climate Zones is an important approach of comprehensively analyzing urban areas of mixed LULC (Bechtel et al., 2019) and may be complementary to our approach of aiming at a separation of the effect of different LULC types.”

Minor comments:

1.5) Line 29: Many studies on satellite-derived UHI, in fact, the bulk of the studies, consider the linear and sometimes non-linear relationships between LST and some proxy for vegetation cover, usually the Normalized Difference Vegetation Index (NDVI). Vegetation cover, particular its cooling effect, does lie on a continuum. By considering tree and treeless green space, the study here focuses on the two points of this distribution. If the reason for doing so is to quantify the magnitude of the effect for these binary cases, which, in my opinion, is less useful for studying the climatic effects of real cities than the continuous distribution used in other studies, based on my first major comment, the authors would have to add sufficient disclaimers to those numbers. A possible way to reframe this is in the context of policy, which the authors mention in the final line of the abstract but do not discuss in detail, is to state that while NDVI is useful for studying the climatic effects of urbanization from the evaporative cooling perspective, it is hard to implement NDVI-based policy solutions. Rather, urban forests versus urban parks are easier to implement, which is why the authors focus on the two points of the continuous distribution.

>>>> Thank you for mentioning this point. The main reasons why we distinguish between trees and treeless green spaces (e.g. parks) is that the two can show very different biogeophysical behavior, particularly during hot extremes (e.g. Shashua-Bar et al., 2009, Teuling et al., 2010, Yosef et al., 2018, Duveiller et al., 2018). For example, the larger root depth of trees may allow water extraction from deeper soil layers and hence lead to larger ET and cooling even during dry and hot extremes. We agree that this behavior may be partly captured when looking at non-linear relationships between NDVI and LST (assuming that urban trees and treeless green spaces have different NDVI values). However, this may still cause some difficulties. For example, it would be difficult to tell whether we observe a certain NDVI within a grid cell, because it is mainly covered by treeless urban green spaces or whether the observed NDVI is a result of a grid cell that is partly covered by trees and partly by non-vegetated areas (assuming that treeless urban green spaces have a lower NDVI than trees and that an averaging of high and low NDVI results in a medium NDVI).

1.6) Line 33: There are studies that look at the temperature of different LU/LC within urban areas [Silva et al. 2018] I would reframe this to the focus on the two primary types of urban vegetation.

>>>> We of course agree that there are several studies that look at the temperature of different LULC within urban areas. Our main point is that the comparison has not been done systematically in different climates/regions. In addition, a consideration of vegetation within cities is often missing. The study of Silva et al. 2018 is an example of a very specific region that has been analyzed and is very interesting. We included a reference to it in the introduction and tried to make clear that we think that different LULC types within or around cities have not been systematically assessed in different regions.

1.7) Line 38: It is true that tree species composition varies across cities. However, the authors never address this in the actual study. I would avoid talking about it here or add a discussion on this based on the kinds of trees one expects in the different regions considered.

>>>> We understand and appreciate this suggestion. We do not explicitly address this in our study, but agree that some readers might expect that we do. Since, reviewer 3 (comment 3.1) suggested to include into the introduction many references to studies (e.g. Rahman et al. (2020)), that also explicitly are about traits of different tree species, we updated this part. However, to make clear that our signals may (implicitly) contain species effect, but that we do not decompose the observed signals in such a way that would allow us to analyze the effect of different species explicitly, we also updated the manuscript.

1.8) Line 130: This is somewhat of an exaggeration. There are multi-city observational studies looking at the effects of urban trees on urban LST [Kroeger et al. 2018]. I would avoid this line.

>>>> We modified the statement in line 130 according to your suggestion. The study of Kroeger et al. (2018) is very interesting. Concerning the heat mitigation aspect, we think that the methodology applied by Kroeger et al. is predominantly based on urban heat island observations and only allows for very indirect estimates of the effect of trees/forests. To try to have more explicit estimates was one of the main aims of our study and hence the study of Kroeger et al. does partly underline our arguments in the introduction. In general, the combination of urban heat island estimates (based on LST, (Imhoff et al., 2010)) and the potential relationship between LST and air temperatures (Zhang et al., 2014) in different biomes, as carried out by Kroeger et al. (2018) is a very important/interesting approach and we added a reference to it.

1.9) Table S2: Landsat is NASA+USGS mission, while Aster is a NASA+METI mission.

>>>> Thank you for this remark. We updated the table accordingly.

References:

Mildrexler, D. J., Zhao, M., & Running, S. W. (2011). A global comparison between station air temperatures and MODIS land surface temperatures reveals the cooling role of forests. *Journal of Geophysical Research: Biogeosciences*, 116(G3).

Novick, K. A., & Katul, G. G. (2020). The Duality of Reforestation Impacts on Surface and Air Temperature. *Journal of Geophysical Research: Biogeosciences*, 125(4), e2019JG005543.

Hu, Y., Hou, M., Jia, G., Zhao, C., Zhen, X., & Xu, Y. (2019).

Comparison of surface and canopy urban heat islands within megacities of eastern China. *ISPRS Journal of Photogrammetry and Remote Sensing*, 156, 160-168.

Zhang, P., Bounoua, L., Imhoff, M. L., Wolfe, R. E., & Thome, K. (2014). Comparison of MODIS land surface temperature and air temperature over the continental USA meteorological stations. *Canadian Journal of Remote Sensing*, 40(2), 110-122.

Chakraborty, T., Hsu, A., Many, D., & Sheriff, G. (2020). A spatially explicit surface urban heat island database for the United States: Characterization, uncertainties, and possible applications. *ISPRS Journal*

of Photogrammetry and Remote Sensing, 168, 74-88.

Zhou, W., Huang, G., & Cadenasso, M. L. (2011). Does spatial configuration matter? Understanding the effects of land cover pattern on land surface temperature in urban landscapes. *Landscape and urban planning*, 102(1), 54-63.

Silva, J. S., da Silva, R. M., & Santos, C. A. G. (2018). Spatiotemporal impact of land use/land cover changes on urban heat islands: A case study of Paço do Lumiar, Brazil. *Building and Environment*, 136, 279-292.

Kroeger, T., McDonald, R. I., Boucher, T., Zhang, P., & Wang, L. (2018). Where the people are: Current trends and future potential targeted investments in urban trees for PM10 and temperature mitigation in 27 US cities. *Landscape and Urban Planning*, 177, 227-240.

Reviewer #2 (Remarks to the Author):

Review for Manuscript#: NCOMMS-20-42315

This is a neat piece of writing. The authors present temperature differences between urban trees, urban non-tree green spaces, and urban fabrics in summer and extreme heat. So, this is a new perspective on integrated urban climate studies, as previous studies have typically compared urban-rural temperature differences. The analysis is mainly based on high-resolution land use/land cover data, Landsat (30m) and ASTER (90m) LST data. The technique for constructing suburban details from satellite observations is impressive.

2.1) In the Methods section, I note that this is based on a Generalized Additive Model, which appears to be a statistical approach. It is important to explain some of the spatial and temporal assumptions for this method. I wondered about the uncertainties regarding these assumptions, especially with respect to LST during the summer day and during these extremely hot days.

>>>> Thank you for this comment. The explanation of the Generalized Additive Models (GAMs) and the involved assumptions is indeed short. Thus, we extended several parts of the Methods section:

“... The x- and y-coordinates are included as two-dimensional tensor product smooths. All other predictors are included in the form of thin plate regression splines. Including spatial coordinates as tensor product smooths reduces spatial auto-correlation and can help to reduce the potentially confounding impact of unobserved phenomena and variables (Beale et al., 2010). Since the structure of GAMs is inherently additive, we may interpret the modelling process in a simplified way: A part of the LST signal is modelled as a function of topographic variables (e.g. elevation) and spatial location (i.e. x-y-

coordinates) and the remaining signal is expressed as a function of the land-cover at a specific location. However, it should be noted that while the effect of the different land-covers is modelled based on smooth functions (i.e. nonlinear functions), we do not model the effect as a spatial interaction term. This means we are interested in the average effect of, e.g., urban trees over the whole city and not in specific patterns within each city. This is justified by the scale of our analysis looking at inter-city differences, but of course intra-city differences can be equally important. “

...

“... We fit a GAM for each LST observation available to be able to distinguish the potential cooling effect of urban vegetation for varying conditions (e.g., varying background temperatures). Since there is a separate GAM for each observation, not only the effect of vegetation on temperature, but the effect of all variables is estimated separately for each observation. ...”

2.2) The description of the LST phenomenon is fairly straightforward. However, explanations of the mechanisms behind the phenomena lack evidence or require a deeper understanding (e.g., in the section on biophysical processes in the discussion, these explanations are possible, but not sufficient). The results may have implications for policymakers, but until the theory of mechanisms is understood, their usefulness for mitigating future climate change may be rather uncertain. In addition, the interpretation of biophysical processes is based on spatial substitution (e.g., L84-88, L192-200). That is, LST phenomena are observed and simulated at the suburban scale, whereas biophysical processes are explained by replacing urban trees with rural forests and urban nontree green spaces with rural pastures. It seems that one should be concerning of this, as their correlation (R^2) in Figs.S1a and b is only about 0.6. Such a hypothesis should be aware of the uncertainty. Also, I wonder if the authors can reconstruct the urban details of albedo, roughness, ET, and other biophysical parameters of urban trees, nontree green spaces, and urban fabrics?

>>>> We agree that biophysical explanations have to be extended and uncertainties need to be further discussed as also suggested by reviewer 3 (comments 3.7, 3.16, 3.21 and 3.22) and in comments 2.3 and 2.5. The spatial substitution is a simplification, which owes itself to the lack of high spatial resolution data on albedo, ET and roughness. We believe that it is to date still extremely difficult to reconstruct the details of these variables at small (neighborhood/street) scales for a large number of cities in different regions. Thus, we now emphasize that additional research is required to better understand biophysical processes within cities in the different regions and that our analysis is limited by this spatial substitution. The correlations between urban trees and rural forests and urban green spaces and rural pastures are very notable, but as you point out they are also related to larger uncertainties. These are now further highlighted in Figure 3 and Figure S 3 and we now dedicate a larger paragraph on the potential differences between vegetation in- and outside of cities (starting in the introduction as also asked for by reviewer 3, comment 3.1).

Added to the main part of the manuscript (please also refer to the response to comment 2.5 and 3.7 for further discussion):

“The variation of environmental conditions along urban-to-rural gradients, which can be very important (Winbourne et al., 2020), seem to have a much smaller impact on the variation in cooling than the variation of environmental conditions across regions. However, several differences between rural and

urban vegetation cooling are noteworthy (Figure 3). The cooling of urban trees in central European regions and particularly in Scandinavia is higher than the one of rural forests. This could indicate that factors potentially contributing to a higher transpiration and cooling rate in cities (e.g. higher background temperatures) outweigh factors that may reduce cooling in cities (e.g. increasing water stress due to insufficient soil volumes). In Turkey the cooling of urban trees is generally much lower than the one provided by rural forests and hence factors reducing the cooling of urban trees in cities may dominate in this region. On the other hand, the cooling of treeless green spaces in Turkey is higher than the one of rural pastures. This could indicate that irrigation of treeless urban green spaces is more relevant than irrigation of urban trees in southern European regions including Turkey. Irrigation may indeed play a relatively small role for urban trees in Europe (Pauleit et al., 2002, Tsiros, 2010). However, such aspects need further investigation and it still seems very difficult to derive a clear picture on urban vs. rural vegetation temperature and transpiration differences. To further validate and elucidate on the urban vs. rural differences of cooling provided by vegetation it will be crucial to generate data on biophysical processes within cities (Chrysoulakis et al., 2018).”

In general, we think that the overall patterns that we see in the temperature differences between urban trees and urban fabric are very robust. However, there are definitely larger uncertainties for differences of the effects between average summertime and hot extreme conditions. To better highlight these uncertainties, we now modified parts of Figure 2 (Figure S 1) including uncertainties in the form of standard errors. We also included additional maps showing the cooling provided by urban trees and treeless urban green spaces during average summertime conditions and hot extremes. These maps show that there is a shift of the largest cooling towards the north, which is in line with what we would expect (assuming that drying during hot extremes in southern European regions will lead to lower ET and temperature differences, whereas in northern regions ET may increase during hot extremes since there may be more energy available and often less soil moisture limitation).

Figure S1: a) Boxplot showing the difference in the cooling provided by urban trees during hot extremes ($\Delta T[K]_{HE}$) and the cooling provided during average summertime conditions ($\Delta T[K]_{JJA}$). Significance levels (0.1-0.05 (.), 0.05-0.01 (*), 0.01-0.001 (**), and 0.001-0 (***)) indicate whether the median is significantly different from zero (based on a Wilcoxon signed-rank test). b) The map shows the difference in the cooling provided by urban trees during hot extremes and the cooling provided during average summertime conditions for each city. In addition, a spatially smoothed trend of these differences is added as a background to the map. The size of the dots indicates the uncertainties in the form of standard errors (SE).

Figure S 17: Comparison of temperature differences between urban vegetation (trees and treeless greenspaces) and continuous urban fabric during average summertime (JJA) conditions and during hot extreme conditions (UT = Urban Trees, UF = continuous Urban Fabric, GS = treeless urban Green Spaces).

Minor comments:

2.3) L61: Why does Gaziantep in Turkey show a higher LST for urban trees than for urban fabrics? Perhaps a note could be added to the discussion.

>>>> We included further discussion regarding this point (please refer to the response to 2.5).

2.4) L58, 70, 72: Do “hot temperature extremes”, “hot days”, and “heat extremes” refer to these orange dots in Fig2b? It would be better to use a consistent statement.

>>>> This is a good point. We now consistently refer to “hot extremes”.

2.5) L65-75: Interesting. Is it because of the stomatal control of trees? Linking this to L61, I wonder what the leaf area index (or soil exposure and canopy openness) is for “urban trees” in Gaziantep. Is it possible that Gaziantep's "urban trees" are evaporating much more than they are transpiring?

>>>> We believe that it could be a combination of factors that needs to be considered here and differences in evaporation and transpiration are likely to be also relevant. We added the following thoughts to the manuscript (also based on comment 2.4):

“It is notable that LSTs can even be higher over urban trees than over continuous urban fabric in southern European regions including Turkey (e.g. in Gaziantep). This may be related to extremely low levels of evapotranspiration over urban tree areas and hence a more significant influence of albedo differences (Wang et al., 2020), that are quite large in these regions. Lower leaf area indices (LAIs) in Mediterranean regions (Iio et al., 2014), could be an additional factor that needs to be considered. If satellites observe a large fraction of dry and even bare soil underneath trees with low LAIs, LSTs may appear to be very high.”

2.6) L76: Should the temperatures in Fig2c mostly be negative in sign, according to the definition in your figure caption.

>>>> Thank you for noticing this. We corrected this, but since Fig 2 changed this does not show anymore.

2.7) For the Results section: It would be better to specify what LST is presented at the section beginning. For example, 24-hr average, daytime average, nighttime average.

>>>> We included a slightly extended methodological explanation (even before) results section as also suggested by reviewer 3 (comment 3.5).

2.8) L86-88: In the main text, it is ET over pastures; in the caption of Fig 4b, it is ET over forests. Which is true? In addition, why R2 in Fig 4b is higher than R2 in Fig 4a?

>>>> It is ET over pastures as correctly indicated in the main text, but not in the figure caption. This has been corrected. We are not entirely sure why the R2 is higher in Fig 4b, i.e., for the correlation of temperature differences between treeless urban green spaces and urban fabric vs. ET over pastures. However, it may be partly explained by the generally lower variation in temperature differences in comparison to the larger variability in temperature differences between urban trees and urban fabric.

2.9) L155: How do you determine the emissivity from remote sensing data for “urban trees”, “green space”, and “urban fabric”?

>>>> We extended the description of how the emissivity is calculated based on (Parastatidis et al., 2017). There are no specific emissivities assigned to the different LULC categories like “urban trees” or “urban green spaces”. It is rather the fraction of vegetation cover as derived from NDVI values that determines the emissivity, which is of course a simplification:

“... We chose NDVI-based emissivity, but also tested the sensitivity of different emissivity sources for a smaller sample of cities (Figure S 8). The estimation of NDVI based emissivities involves three steps (Parastatidis et al., 2017). First, NDVI is calculated for each grid cell based on Landsat observations.

Second, relying on an empirical relationship, the fraction of vegetation cover (FVC) is calculated based on NDVI values (Carlson and Ripley, 1997). Third, the emissivity is calculate based on FVC assuming that non-vegetated surfaces have an emissivity of 0.97, vegetated surfaces have an emissivity of 0.99 and all partly vegetated surfaces are a linear combination of these two emissivities and hence lie between 0.97 and 0.99.”

2.10) L166-170: Spatially averaging or temporally averaging of LST? Also, these two sentences are difficult to understand precisely. Could the author have made it clearer?

>>>> We reformulated the sentences and emphasize that we mean “temporally” averaging LST:

“These findings emphasize that temporally averaging LST observations before deriving the impacts of vegetation on temperature may obscure the cooling potential when it may be most important (i.e. during hot extremes). It also shows that in some cities the average reduction of LSTs in summer is lower than during hot temperature extremes, but in other cities the opposite is the case. This may also foster the debate whether high cooling during a short hot period is more relevant than a high cooling during longer less extreme periods when it comes to mitigating the adverse impacts of urban heat. In particular, this could be relevant when comparing different heat mitigation strategies that may also have a reduced or increased effect during hot extremes. “

Reviewer #3 (Remarks to the Author):

Review of “The Role of Urban Trees in Mitigating Heat in European Cities” by Schwaab et al. (NCOMMS-20-42315).

The authors investigate the cooling effect of urban trees in 293 European cities combining high resolution Land Surface Temperature (LST) observations with land use/land cover (LULC) data and a detailed Street Tree layer from the Copernicus Urban Atlas. They find that, during heat extremes, urban trees are 7-10 K cooler than urban surfaces in central Europe but only 0-4 K cooler in southern regions (0-4 K). They also show that LST differences between urban trees and treeless green spaces can reach 6 K. This study provides novel insights on the cooling effect of urban greening during average versus extreme temperatures and, especially, on the differences between tree and treeless urban green spaces. However, I have some concerns on the methodology/quality of the paper (see comments below) and, in my opinion, the manuscript needs significant improvements before it can be considered for publication in Nature Communications.

General Comments

First of all, the manuscript structure/content should be improved. Specifically:

3.1) The introduction is too succinct and does not provide a sufficient overview of the existing literature on the effects of vegetation on urban climate (see, for example, Wang et al. (2018, 2019), Rahman et al.

(2020), Winbourne et al. (2020) and references therein). I think the manuscript would benefit from a brief (but detailed) discussion on key processes and main results in the literature (e.g. shading/evapotranspiration, observed air/surface temperature changes).

>>>> Thank you for this comment. We modified the introduction and included a more detailed discussion on key processes:

“Trees influence urban climate primarily via shading and transpiration (Winbourne et al., 2020). Shading can strongly reduce daytime LSTs and air temperatures (Wang et al., 2018), with the effect usually being larger over asphalt than over grass surfaces (Rahman et al., 2019, Rahman et al., 2020) and being larger in shallow than in deep street canyons (Coutts et al., 2016). The shading effect depends, amongst other factors, on mainly the morphological characteristics of different trees/tree species and has been shown to increase with LAI (Smithers et al., 2018, Rahman et al., 2020). The amount of transpiration and its effect on temperatures depends on characteristics of trees/tree species, but is also strongly dependent on environmental conditions that have, e.g., an influence on stomatal conductance of trees (Winbourne et al., 2020). The environmental conditions that influence transpiration of trees and their potential to reduce temperatures are changing, e.g., for different seasons, during extreme conditions, with different geographical contexts and along gradients of urbanization (Wang et al., 2019a, Su et al., 2020, Meili et al., 2021).

Seasonality has a strong influence on the cooling potential of vegetation (Manoli et al., 2020a, Su et al., 2020). In many regions, temperature differences between vegetation and urban fabric are larger during summer than during winter (Su et al., 2020). However, in dry regions including parts of southern Europe, the summertime cooling provided by vegetation can be reduced due to limited soil moisture limited ET (Manoli et al., 2020a). As results for the US show, the cooling provided by urban trees during cold extremes is much smaller than during heat extremes and the amount of transpiration can be closely connected to the variation in saturated vapor pressure (Wang et al., 2019a). The two opposing effects of an increased surface resistance during hot extremes (due to soil moisture limitation and stomatal behavior) and an increased vapor pressure deficit (mainly due to increased temperatures) can either lead to an increase or a decrease in temperatures over vegetation during heatwaves (Wang et al., 2019b). However, our understanding is still limited of how temperatures respond to these contrasting effects in different geographical and climatic contexts.

The potential of trees to reduce temperatures via transpiration is influenced by the characteristics of the surrounding land and may be different for trees within a city in comparison to trees or forests in rural areas (Pataki et al., 2011, Mussetti et al., 2020). The environmental conditions in an urban surrounding could either increase or decrease the temperature reduction caused by trees (Winbourne et al., 2020, Czaja et al., 2020). For example, high CO₂ concentrations (Brondfield et al., 2012), increased nutrient availability (Decina et al., 2017), high temperatures (Zipper et al., 2017) and high levels of irrigation (Gao and Santamouris, 2019, Reyes-Paecke et al., 2019) may regularly be encountered in cities and can increase transpiration and cooling (Melaas et al., 2016). On the other hand, several factors that may negatively affect growth of trees and their cooling effect need to be taken into account (Chen et al., 2015). High temperatures in cities can increase water stress (Meineke et al., 2016), insufficient soil volumes and soil compaction is limiting root growth (Jim, 2019) and increased air pollution can have many additional adverse effects (Chen et al., 2015). Due to the different environmental influences and

tree species in cities, it is not clear whether studying rural forests allow us to draw conclusions on the cooling potential of trees within urban areas. “

3.2) The article has no conclusions – it ends with a “limitations” section and the reader is left wondering what the take-home message is.

>>>> We rearranged the Discussion section and modified it so that there is a paragraph at the end that resembles a conclusion section. We chose this compromise since the format of Nature Communications allows for a Discussion section at the end, but not for a separate conclusions section.

The conclusion part within the discussion now reads like this:

“We present an observation-based analysis of temperature differences between urban trees and urban fabric across European cities. Combining high spatial resolution LST and LULC data within a large number of cities, the presented results are not derived from urban heat island data, which are often the basis for analysing the cooling potential of trees in different regions. Using high-resolution data at intra- and inter-city scales enables us to show that the potential cooling benefits depend on vegetation type as well as on climatic context. In general, urban trees are related to reductions in LSTs that are 2-4 times higher than the LST reduction related to treeless urban green spaces. Both types of vegetation lead to a high reduction of LSTs in Central Europe and a smaller reduction in Southern Europe. While urban trees and rural forests predominantly provide cooling in all European regions, treeless urban green spaces and rural pastures exhibit a very small cooling benefit or even a warming effect in Southern European regions.

Even though vegetation within urban areas is subject to different environmental conditions and human influence than vegetation outside of cities, the cooling provided by rural vegetation and urban vegetation shows similar regional patterns in Europe. These patterns are closely related to differing evapotranspiration rates across regions. In addition to regional variations, there are also substantial seasonal variations in the cooling provided by urban trees and there is a notable influence of hot extremes. The LST reduction during hot extremes decreases in the Mediterranean and in the Iberian Peninsula, but rather increases in Scandinavia and the British Isles. In summary, our results confirm the high potential of trees to mitigate urban heat in Europe and highlight important spatio-temporal variations in their cooling effect.”

3.3) In general, the language should be improved (there are several repetitions, see specific comments below).

>>>> We removed repetitions (also following your specific comments). In addition, the manuscript has been sent out for professional language editing.

3.4) One of the most interesting results here is the quantification of tree cooling versus the effect of other green spaces. However, the manuscript falls short in providing some key information thus

diminishing the overall quality of the results/discussion. Specifically:

The authors should clarify in the main text how the different land cover types have been defined. Do “street trees” include urban parks, urban forests, etc? What is considered “green spaces”? Are street trees included in the “green spaces” or not? What about forests and pastures? For example, do pastures include rural crops, grass, etc? I think the manuscript would benefit from some clarifications (e.g. distinguishing between trees, grass, shrubs or considering LAI). The authors often refer to “different vegetation types” (e.g. line 2018) but these are not clearly defined.

>>>> This is an important point and we agree that clarification is necessary on how the different land-use/land-cover (LULC) types within each city can be defined. We now added a new table to the Appendix. In addition, we added a description/discussion in the text to clarify that urban green spaces (or green urban areas) as defined by Copernicus can also include trees. We calculated the fraction of green spaces covered by trees for every city, which is on average 28%. However, fitting a regression model to the fraction of trees and green spaces allows to separate the signals of trees and green spaces without trees. We also tested this by removing areas - that are defined as green spaces and at the same time covered by urban trees - from the analysis to understand if this would change the signal that we obtain for green spaces. The effect was negligible and we think it is in general more useful to integrate all data and separate the signals with the help of the defined LULC fractions per each grid cell. However, the signal of urban trees above vegetation can of course be different than the signals above paved areas (Hardin and Jensen, 2007, Gillner et al., 2015). We did not explicitly distinguish between these two signals for each city, but think that this could be very interesting in future studies, since this effect will most likely be region-dependent.

Table S 5: Definition of the LULC types that were used to calculate LST differences (adapted from Copernicus (2016)).

LULC type	Description
Continuous urban fabric	Land cover: Degree of soil sealing > 80% Built-up areas and their associated land. Buildings, roads and sealed areas cover most of the area; non-linear areas of vegetation and bare soil are exceptional Land use: Predominant residential use: areas with a high degree of soil sealing, independent of their housing scheme (single family houses or high rise dwellings, city centers or suburb).
Urban trees/Street tree layer	The Street Tree Layer (STL) includes contiguous rows or patches of trees covering 500m² or more and with a minimum width (MinMW) of 10 m over “Artificial surfaces” (nomenclature class 1 of the urban atlas) inside urban areas (i.e. rows of trees along the road network outside urban areas or forest adjacent to urban areas should not be included).
Urban green spaces/Green urban areas	Public green areas for predominantly recreational use such as gardens, zoos, parks, castle parks and cemeteries. Suburban natural areas that have become and are managed as urban parks. Forests or green areas extending from the surroundings into urban areas are mapped as green urban areas when at least

	two sides are bordered by urban areas and structures, and traces of recreational use are visible.
Pastures	Pasture and meadow under agricultural use, grazed or mechanically harvested. Wooded meadows. (Not included are fields under crop rotation systems).
Forests	Broad-leaved forest, coniferous forest and mixed forest; Transitional woodland and shrub (clear cut, new plantations and regeneration, or damage forest); With ground coverage of tree canopy > 30%, tree height > 5 m, including bushes and shrubs at the fringe of the forest; Included are plantations such as Populus plantations, Christmas tree plantations; Forest regeneration / re-colonization: clear cuts, new forest plantations. Not included are: Forests within urban areas and/or subject to high human pressure.

3.5) The authors should clarify (i.e. using the same symbols and terminology) which variables are related to “trees” and which ones are related to “other green spaces”. For example, in Fig 1 it would be useful to replace DeltaT with DeltaT_trees and DeltaT_green or use the same subscripts employed in Figure S1-S2. It should also be clarified in the main text how temperature differences are computed. In other words, do they represent the difference between the average temperature of all the “tree/green” pixels and the average of all the “continuous urban fabric” pixels (i.e. one value per city) or is DeltaT a spatially distributed variable (as, for example, in the Global SUHI Explorer, <https://yceo.users.earthengine.app/view/uhimap>)? This is partly addressed in the Methods section but it is not fully clear. For example, does the “spatial variation of the LST differences between vegetated urban land and urban fabric” in Line 360 refer to intra-urban or regional (city-to-city) variations? Some clarifications are needed to help the reader understand the methodology upfront (i.e. before the results section).

>>>> Thank you for the suggestion to clarify the labels. We changed all labels to indicate whether the ΔT refers to differences in temperature between urban trees and urban fabric or other temperature differences. In addition, we added clarification to the Methods section, that we do not analyze intra-urban LST variations and refer to regional (city-to-city or inter-city) variations when discussing spatial variation. As suggested we also added some more methodological explanation before the results section (please also refer to comment 2.7) and we now try to make clearer that the temperature differences between different LULC types are based a statistical model (Generalized Additive Model) fitted to LST observations. They are not simply the average of all grid-cells that are, e.g., either classified as urban trees or as continuous urban fabric since such an approach usually tends to weaken signals (due to mixed pixels).

3.6) I think there are some missed opportunities here. For example, an overview of the percentage of tree cover/green spaces in the analysed cities would be quite interesting and straightforward to show. In general, results are shown for different cities/regions/countries, which is interesting but limits the generalizability of the conclusions. What about possible trends with urban or background climate characteristics (see specific comments below)?

>>>> We calculated the percentage of artificial urban areas (as a proxy of the size/area of each city, Table S 3) being either covered by trees or by green spaces. We included this information in the

Appendix. However, we think that this is not having a major influence on our results. As also explained in the response to comment 3.5 we calculate the effect of urban trees/urban green spaces by using the fraction of trees/green spaces in each grid cell to fit a statistical model. This model will be rather independent of the total amount of trees and green spaces within a city (in contrast to more classical ways of calculating the SUHI, which would of course be dependent on how much vegetated areas we find within each city or the area that is defined as urban). On the other hand, we understand and agree that at some point the biophysical processes that are related to cooling will change depending on the amount of trees (as also partly discussed in the response to comment 1.4). For example, humidity and temperature are of course influenced by the amount of trees and green spaces in the neighborhood and a large percentage of trees in a city will have a different influence than a small percentage, when it, e.g., comes to feedbacks concerning the ET of trees (i.e., higher humidity and lower temperatures due to more trees could lead to less ET etc.).

Figure S 20: Fraction of artificial urban surfaces covered by trees and fraction of artificial urban surfaces covered by green spaces.

3.7) A clear discussion on the key biophysical mechanisms (e.g. ET, water stress, albedo, roughness) causing the observed changes in LST is somewhat missing. This is partly addressed at the end of the discussion section but the arguments are rather qualitative and it is unclear whether they refer to city-scale or street canyon conditions (e.g. at the canyon level urban trees can increase or decrease roughness depending on building height, while at the city level the impact of trees on urban roughness is probably negligible).

>>>> We extended the discussion on biophysical mechanisms (also according to comment 2.2. and 3.14). We agree that the discussion of roughness effects can be improved and added a modified paragraph (please refer to comment 3.22).

Added and modified paragraphs in the manuscript:

“The lowest temperature differences between urban trees and urban fabric are observed in cities in southern European regions and are related to low evapotranspiration rates (Figure 4), which can be linked to increased surface resistance due to limited soil moisture availability (Wang et al., 2019b, Denissen et al., 2020). High temperatures during summertime in the Mediterranean and during hot extremes have the potential to increase ET through high vapor pressure deficit (Wang et al., 2019b, Meili et al., 2021). However, transpirational cooling of trees often decreases considerably due to reduced stomatal conductance (McAdam and Brodribb, 2015). Certain tree species keep their stomata open even during hot extremes, possibly to create a cooling effect through transpiration (Teskey et al., 2015). Hence, there are regions in which trees show an increase in transpiration during hot extremes (De Kauwe et al., 2019, Harrison et al., 2020). The species-specific response to high temperatures and drought conditions (Roman et al., 2015) overlays the effect of environmental conditions (e.g., amount of soil moisture) in ways that are not directly captured in the MODIS ET product and cannot easily be disentangled. Since the cooling of urban trees during hot extremes shifts north and increases over the British Isles, Scandinavia and parts of Mid-Europe/Alps, we assume that higher VPD causes an increase in transpiration in those regions and a decrease in the Mediterranean and Turkey, where we see a decrease in cooling during hot extremes.

In comparison to ET, albedo seems to play a minor role in explaining the inter-city temperature differences between urban trees and urban fabric. However, while inter-city differences may not be strongly influenced by albedo, the temperature differences between urban trees and urban fabric in specific regions most likely are. In particular, the albedo may have a larger effect in dryer areas such as Southern Europe (Wang et al., 2020), and it may increase during hot extremes that are associated with large amounts of incoming shortwave radiation (Davin et al., 2014, Perkins, 2015). It is notable that LSTs may be even higher over urban trees than over continuous urban areas in Southern European regions and Turkey (e.g., in Gaziantep). This may be related to extremely low levels of evapotranspiration over urban tree areas and hence a more significant influence of the high albedo of urban areas in Southern Europe. Lower leaf area indices (LAIs) in Mediterranean regions (Iio et al., 2014) could be an additional factor to be considered. If satellites observe a large fraction of dry and even bare soil underneath trees with low LAIs, LSTs may appear to be very high.”

3.8) My major concern is about the use of DEM/Aspect data from Copernicus. Are they representative for cities? I am not familiar with the details of this dataset but I doubt that it includes information on urban morphology (which is quite complex at the resolution considered here). Global scale datasets of building heights and 3D urban structures are now available (e.g. Li et al. 2020) and, in my opinion, should have been employed here. In other words, topography can play a role in a city like Zurich but, in general, the intra-urban variability of LST is largely controlled by the 3D structure of the urban fabric and its impact on the surface energy fluxes. These are key considerations that seem to be overlooked. It is also not fully clear to me how ET and albedo data have been used (see specific comments below).

>>>> This is an important comment and we thank you for the suggestions. The DEM data does not include the 3D structure of buildings. We included DEM data as well as the aspect information, because

(as you mention) there are topographically diverse cities (such as Zurich) where urban trees may be, e.g., more frequent in higher/lower elevations or on southern/northern slopes etc. Including the DEM and aspect data avoids that the effects of these two variables are falsely attributed to the land-cover (e.g. amount of urban trees). We agree that the 3D building structure will also have an effect. However, as has been shown in other studies (Logan et al., 2020, Wu et al., 2020), this effect might be relatively small concerning LST differences in comparison to the effect of soil sealing and land cover type (the 3d building structure effect is likely to be more relevant for air temperatures). The results of Logan et al. (2020) and Wu et al. (2020) may seem surprising at first since anisotropy effects tend to be significantly depending on 3D urban form (Krayenhoff and Voogt, 2016). However, the use of Landsat (as used in both studies), which has observation angles approximately at nadir, may alleviate anisotropy effects (Bechtel et al., 2019).

Despite some of this literature-based evidence, we tested the potential effect of 3D urban structure looking at two different datasets. First, we included the data provided by Li et al. (2020). However, the resolution of the data is 1000 m and hence was difficult to analyze in a useful way together with our data at 90 m resolution. In a second approach, we included data from Copernicus on building height (<https://land.copernicus.eu/local/urban-atlas/building-height-2012>). This data includes the building height for each grid cell on a 10 m resolution. A disadvantage in comparison to the data of Li et al. 2020 is that it is only available for one city (i.e. the capital city) in each European country and it is only available for a relatively small fraction of each city. Yet, we were able to show that the building height is closely related to the different types of urban fabric (i.e. the amount of soil sealing) in the Copernicus urban atlas (Figure S 15). Thus, including the building height as additional variable did not change our results substantially as we were able to show when rerunning all models for the cities in which building height was available (Figure S 16). However, neither including data on different urban fabric categories (i.e. soil sealing) nor on building height (i.e. 3D structure) would change the results and hence at least one of the two should be included.

We also believe that an analysis of the potential temperature differences between urban trees and urban fabric at different locations within a city (intra-city analysis), will depend to a much larger degree on the 3D structure than an inter-city analysis. As has been shown, e.g., street trees at the bottom of a deep street canyon may have less of a cooling effect, since the street is already partly shielded from direct incoming solar radiation (e.g. Coutts et al., 2016). It could be very interesting to analyze whether these effects would vary with different background climates. For example, it may be possible that street canyon properties have more of an effect on temperature differences (between trees and urban fabric) in regions where there is more incoming solar radiation.

Figure S 15: Boxplots of building heights for each category of urban fabric in the Copernicus urban atlas. Dense urban fabric categories are associated with higher buildings.

Figure S 16: Correlation between temperature differences (urban trees – minus continuous urban fabric), when building height is included as an additional predictor variable. When building height is included in the model fit, the prediction of temperature differences between urban trees and continuous urban fabric includes average values of building height in areas where we find continuous urban fabric and a building height of zero for urban tree areas.

Specific comments (reviewer 3)

3.9) Line 16: “different pattern”, explain better

>>>> We reformulated the sentence in the manuscript. It now reads:

“Absolute LST differences between urban trees and treeless urban green spaces are particularly lower than absolute temperature differences between urban trees and urban fabric in central and northern European regions, but not in southern European regions.”

3.10) Lines 25-28: repetition “Based on”. Also, I guess this part should be on “urban vegetation”, thus justifying the following sentence on the lack of studies distinguishing between different vegetation types (e.g. trees vs no trees).

>>>> Thank you for spotting the repetition. We removed the second “Based on” and slightly reformulated the sentence.

We still would like to emphasize in this part of the introduction the “urban tree” aspect, since we focus mainly on comparisons between urban trees and urban fabric. However, we agree that it would almost

fit better to the next sentence if we referred to “urban vegetation”. As a compromise, we changed it to “... the magnitude by which urban trees and other urban vegetation may reduce urban heat...”

3.11) Line 29, 32, etc: language can be improved.

>>>> We reformulated this sentence:

... . Since SUHIs are usually estimated as the differences in land surface temperature between cities and their surroundings, it can be difficult to distinguish between the effect of different types of vegetation (e.g. urban trees vs. treeless urban green spaces) on temperature...”

3.12) Lines 39-40: “urban areas” is repeated 4 times here. Please rephrase to improve readability.

>>>> Several parts of the introduction (including line 39-40) have been reformulated. Thus, the repetition of “urban areas” does not occur anymore in the revised version of the manuscript. Please also refer to comment 3.1.

3.13) Lines 71-73: “is clearly lower ... In contrast, the difference (...) is low”, this sentence is unclear. Please clarify.

>>>> The results section has been updated and several sentences reformulated. Instead of referring to “low temperature differences”, we now try to better explain that in certain regions there is hardly a difference in the cooling provided by trees when comparing average summertime and hot extreme conditions.

3.14) Line 85-88: what about the correlation between LST differences of urban trees and the ET of rural pasture and the LST of urban green spaces and the ET of rural forests? They might be correlated as well (see other comments below). Actually, if possible, it would be interesting to compare the characteristics of urban and rural vegetation (e.g. considering LST or albedo given that ET is not available for urban areas here).

>>>> Our response to this comment is complemented by the response to comment 3.20 and 3.35. The LST difference of urban trees minus urban fabric is of course also correlated to the ET of rural pasture. As we also explain in comment 1.5 we think that urban trees are closer to rural forests in their biophysical behavior/impact and that treeless urban green spaces are closer to pastures. However, we are of course aware that this is only an approximation and that urban trees within a city are often characterized by different species and are influenced by different environmental conditions than trees/forests in rural areas. Similarly, pastures can certainly not be directly compared to treeless green spaces which are usually urban park areas characterized by some form of lawn or grassland.

We also agree that it is interesting to show the LST differences between urban fabric and rural vegetation. Thus we included them in Figure 3 and in maps that show the smoothed differences in temperature between urban fabric and forests and urban fabric and pastures (see also comment 3.17, Figure S 17). Accordingly, we also discuss the differences between the ΔT s of urban fabric and urban vegetation and the ΔT s of urban fabric and rural vegetation. It seems that the cooling provided by urban trees in comparison to rural forests is higher in some parts of central and northern Europe. A possible reason amongst others could be that higher temperatures in urban areas in these regions increase ET of urban trees. On the other hand, increased temperatures can lead to increased loss of soil moisture and a decrease in ET. This may be compensated by higher levels of irrigation in urban areas. It remains challenging to disentangle the many different interacting effects of an urban and rural environment on the cooling potential of urban and rural trees/forests. Being able to better attribute differences in the cooling provided by trees/forests within and outside cities could be very helpful. For example, such knowledge could be used to quantify different levels of irrigation within and outside of cities and to understand how we can increase the cooling provided by trees without making them more prone to drought- and heat-induced damage.

3.15) Line 134: typo, “that that”

>>>> Thank you for spotting this. It has been corrected.

3.16) Lines 143-145: this is quite obvious – the cooling effect of urban green spaces (even if treeless) is clearly different from that of urban fabric.

>>>> Yes, we agree that this is quite obvious. What we had in mind was to emphasize that temperature differences between urban trees and urban fabric and temperature differences between urban trees and treeless urban green spaces show different patterns/trends. Since this is also discussed later on, we removed the part “and depends on whether we estimate the potential cooling of urban trees in comparison to treeless urban green spaces or in comparison to continuous urban fabric.”

3.17) Lines 146-149: this is interesting – is it shown somewhere?

>>>> Lines 146-149 refer to the finding that the absolute temperature differences between urban trees and treeless urban green spaces can be higher than the ones between urban trees and urban fabric in some European cities (indicating that urban fabric can be cooler than urban green spaces). This is rather indirectly shown in the boxplot in the main section. Thus, we now show the temperature differences for four cases 1.) urban trees – urban fabric 2.) forest – urban fabric 3.) urban greenspaces – urban fabric 4.) pastures – urban fabric, as additional maps in the appendix. We also discuss what could be the reason why urban fabric might be under certain circumstances cooler than urban green spaces which may of course also be very closely related to the oasis effect (or the urban cool island effect) that is observed in dry regions.

Figure S 3: Smoothed temperature differences between urban trees and urban fabric (UT-UF), forests and urban fabric (F-UF), green spaces and urban fabric (GS-UF) and pastures minus urban fabric (P-UF).

3.18) Lines 168-170: this is also very interesting – some references/information on this debate would be useful (i.e. is there any evidence supporting one or the other viewpoint?)

Line 168-170 was: These findings emphasize that averaging LST observations before deriving the impacts of vegetation on temperature may obscure the cooling potential when it may be most important (i.e. during hot extremes). This may also foster the debate whether high cooling during a short hot period is more relevant than a high cooling during longer less extreme periods when it comes to mitigating the adverse impacts of urban heat.

>>>> Unfortunately we could not find any literature that engages in the debate whether a high cooling during a short hot period is more important than a high cooling during longer less extreme periods. But we think that our results show that this could be an important point that should be kept in mind.

3.19) Lines 174-175: “water scarcity”, what about the projections for the regions where trees seem to be more important? A discussion or, at least, a reference would help.

>>>> Recent results (Christidis and Scott, 2021) show that many European regions could experience drying. We updated the discussion accordingly:

“Projected drying in European summers in these regions are likely to further reduce vegetation benefits (Hari et al., 2020, Christidis and Stott, 2021). However, drying may not only occur in southern Europe, but in many European regions (Christidis and Stott, 2021). Hence, we may see a decrease in cooling even in regions where we presently see the highest cooling.”

3.20) Line 182: is it associated with ET of rural forests or ET of urban trees? What is generating urban cooling is the ET of urban trees which should be comparable with that of rural forests. However, UHI are generally defined as urban-rural differences, so that changes in rural ET can also modify their magnitude. Please clarify.

>>>> This is an important comment. We now added a new paragraph as also discussed in our response to other comments (comment 2.2, 3.14 and 3.35). As you mention the cooling provided by urban trees is influenced by the ET of urban tree areas. Unfortunately, it is extremely difficult to obtain proper estimates of ET of urban vegetation in different regions and cities. A comparison of the urban tree cooling with the ET estimates of rural forests is not ideal, but it still shows that many environmental variables are likely to be similar within and outside of the city (e.g. similar amounts of precipitation, similar background temperatures etc.) and hence there is a correlation between the rural ET estimates and the urban tree cooling.

3.21) Line 184-187: this is not a hypothesis, impermeable urban surfaces have negligible ET. This and the following discussion are somewhat obvious (or just unclear to me).

>>>> Thank you for highlighting this. This sentence was not very well written. We now clarified that we want to express that the LULC category that is defined as “continuous urban fabric” seems to have very low ET. This not exactly surprising either as it is defined as mainly sealed/impermeable surfaces. However, it is nice that our results seem to be in line with this. The sentence now reads:

“... Thus, ET over continuous urban fabric seems to have a minor effect, which was expected since areas defined as continuous urban fabric are mainly sealed and impermeable surfaces.”

3.22) Lines 204-209: the discussion on roughness is unclear. For example, does “roughness of forests” (line 205) refer to urban or rural forests? In cities, given the limited spatial extent of green spaces, roughness is largely controlled by the urban fabric. Also, there are many studies on the subject that should be considered(e.g. Zhao et al. 2014, Li et al. 2019).

>>>> The discussion on roughness effects was indeed not very precise. We updated it in the following way:

“Substantial temperature differences between tree-covered areas and green-spaces and between rural forests and rural pastures in several parts of Europe may be explained by differences in evapotranspiration and surface roughness (Teuling et al., 2010, Yosef et al., 2018, Burakowski et al., 2018, Duveiller et al., 2018). For example, trees are associated with a larger root depth (Schenk and Jackson, 2002) that allows higher exploitation of soil moisture and sustaining larger evapotranspiration rates when the upper soil layers are dry (Yosef et al., 2018). Rural trees and forests typically exhibit a high surface roughness increasing the efficiency of heat convection and may therefore also be an

important factor explaining the high temperature differences between rural forests and rural pastures in Southern European regions (Rotenberg and Yakir, 2010). For large patches of urban trees and treeless urban green spaces similar roughness effects as for their rural counterparts (i.e. rural forests and rural pastures) may be relevant. However, surface roughness of vegetated areas usually interacts in complex ways with the surrounding urban structure. Trees within street canyons can decrease the roughness and leading to reduced turbulent exchange particularly if trees are smaller than surrounding buildings (Meili et al., 2021). If they are higher, they can also increase roughness (Giometto et al., 2017). Roughness effects can also be important for an explanation of the urban heat island magnitude in different regions since the surrounding of urban areas may convect heat more (wet climates) or less (dry climates) efficiently than urban areas (Zhao et al., 2014). However, more recent results suggest that the effect of aerodynamic resistance (mainly controlled by surface roughness) is less relevant for explaining the spatial variation of urban heat islands than the imperviousness that controls evapotranspiration (Li et al., 2019).”

3.23) Lines 235-236: I agree – why this was not included in the analysis? See previous comment on the availability of building height datasets.

>>>> According to your previous comment we now included an analysis using data on building height for nearly 30 cities. Please refer to comment 3.8 for more details.

3.24) Line 254: what is the reason for selecting cities by creating a regular grid?

>>>> The idea behind using a regular grid was to select a city from each part of Europe for the analysis, in order to have a good representation of cities (and the different cooling signals) in different geographical and climatic contexts.

3.25) Lines 265-275: do topography/aspect calculations refer to natural or urban surfaces? How is this information used? I guess the complex 3D structure of the urban fabric is not accounted for so I am not sure how topography and aspect can be estimated for urban areas at a 10m resolution (see general comments above).

>>>> As also discussed in comment 3.8, we included topographic variables such as elevation in order to account for potential confounding factors. However, the elevation data that we included does not in any way account for more complex 3D structures in the city. To better understand whether this would have an effect, we included additional data which shows that the effect of the 3D structure may not substantially influence the results of inter-city comparisons, but will very likely have consequences when looking at LST patterns within each city (see comment 3.8 for more details).

3.26) Lines 312-313: Is the “contribution of different LULC types to the observed ET and albedo values” illustrated somewhere?

>>>> We illustrated this contribution for the albedo values in Figure S 4 for forests and continuous urban fabric to show regional variation. We replicated this figure for ET values (Figure S 5) showing the contribution of rural forests and rural pastures to ET (since ET for urban areas is not available).

Figure S 5: a) Evapotranspiration (ET) estimates over forests for cities in different regions. b) Evapotranspiration (ET) estimates over pastures for cities in different regions. Black dots indicate the mean ET values in each region.

3.27) Lines 316-317: why did the model provide negative values? How reliable are the results for the other cities? In general, it should be clarified how the regression models were produced.

>>>> Negative values occurred in three cities in Turkey (including, e.g., the city Kars). Since the resolution of MODIS ET is 1 km, this meant that there are small cities for which there are not many observations (i.e. LULC pixels) available. In addition, it can happen that not all LULC categories are well represented (this is, e.g., the case in Kars where the amount of forests is low). With only a small number of pixels, the ET estimates are more uncertain and it is possible that, e.g., collinearities may have an effect on the regression coefficients and hence negative coefficients can occur. We added a discussion to highlight these uncertainties. However, for a large majority of cities there is generally enough LULC data available to consistently estimate ET for different LULC categories.

We also added more explanation on the models: “...To estimate the contribution of different LULC types to the observed ET and albedo values, we fitted multiple linear regression models using the fraction of each LULC type as predictor. We used the same predictors as for the models to predict LSTs (Table S 3), but the LULC fractions were calculated for the spatial resolution of Modis ET and albedo. We included all predictors in the form of linear terms. ...”

3.28) Line 337: define R2

>>>> With R2 we mean the coefficient of determination. We added:

“... All fitted models showed a decent coefficient of determination (R^2) ...”

3.29) Line 346: how is “background temperature” defined? Is it the “spatial mean LST of each satellite observation” (line 355)? I guess this might be affected by the fraction of urbanized area of the Landsat image

>>>> Yes, the background temperature was defined as “the spatial mean LST”. As also pointed out by reviewer 1 (comment 1.2), this way of calculating the background temperature could indeed be affected by the fraction of urbanized area of a Landsat LST image. Thus, we changed this definition of background temperature (for more details please refer to comment 1.2) and included a new dataset (E-OBS) that allows to estimate the background temperature independently of LULC effects. The method section has been adapted accordingly.

3.30) Line 582: typo, “large”

>>>> Thank you for spotting this. We corrected this.

3.31) Figure 1: The figure is quite catchy but not fully explicative to me. For example, given that the stacked bars are illustrated without the y-axis, the magnitude of the error bars is not directly quantifiable. Also, the map is very interesting but little visible. And what is B & H? Please revise and/or clarify.

>>>> We added labels to the stacked bars so that the magnitude of the error bars is a bit easier to quantify. We also slightly increased the size of the map in relation to the bars around it, so that the map is more visible. In addition, we added the smoothed map of the temperature differences between urban trees and urban fabric to the Appendix (together with maps of spatially smoothed temperature differences between urban fabric and either forest, pastures or treeless urban green spaces, see comment 3.17). B & H stands for “Bosnia and Herzegovina”. We added clarification to the figure caption.

3.32) Figure 2: how were the 22 cities selected? What is the ordering criteria for the x-axis in panel b? Why not showing an urban or climatic variable there (e.g. urban area, background temperature or similar)? In panel c, a different colorbar would help (the transition from negative to positive values is not clearly distinguishable). And what about the cities in panel d? Are they exemplary cities? It might be useful to highlight them on the map (e.g. in panel C). Also, please write Athens (and not Athina).

>>>> We chose cities that are geographically well-distributed over the whole study area. Ultimately, the choice was slightly arbitrary. To improve this, we now ranked cities according to how close they are to 1. average cooling effect 2. cooling during hot extremes and 3. difference between cooling during hot extremes and average summertime conditions in different regions. From the three highest ranked cities, we selected the one with the highest population. These steps guarantee that the selected cities represent each region well. In addition, we included Gaziantep (Turkey) as an example of a city where

urban trees show higher LSTs than urban fabric areas. The idea of including examples of some cities is to make the analysis less abstract to the reader by referring to actual cities (by their names). The interpretation of the figure seems easiest if the ordering is according to the median cooling in each region. Concerning the map, we replaced the colorbar with one that shows a clear change around zero (from red to blue). We now used the same cities in panel c (formerly panel d) to show seasonal differences. We also replaced the city names with English names (even though Athens has anyway been removed due to the new selection of cities).

3.33) Figure 4: I guess there is a typo in the legend, line 124 – is ET in panel b for forests or pastures (see main text)? Maybe, clarify this also on the axis (e.g. ET_forest, ET_pasture and DeltaT_trees, DeltaT_green).

>>>> Thank you for spotting this. We corrected the typo. We also made the axis labels more explicit (also in accordance to comment 3.5).

3.34) Figure 5: this is very nice. Does the DeltaT-T relation always have that shape? Is the relation based on data or is it a conceptual representation? If it is a conceptual diagram, it would be useful to see some examples of “real data” (e.g. as additional figures in the SI).

>>>> This is a conceptual representation. We included additional figures with examples of “real data” of European capital cities into the Appendix (Figure S 18). The conceptual diagram shows a $\Delta T - T$ relation that is rather similar to the shape in the Mediterranean. In addition, we added confidence intervals to the plot, since we are now trying to more consistently include uncertainty into the figures (cf. Figure 2 and Figure S 1).

Figure S18: LST differences between urban trees and continuous urban fabric for different background temperatures shown exemplarily for 20 randomly selected European capitals. The blue lines indicate a smooth function that is fitted to approximate these differences based on background temperature. Uncertainties are indicated in the form of a confidence interval.

3.35) Figure S1 (now Figure S2): what about the correlation between green spaces and forest as well as urban trees and pasture? I don't think that they will be uncorrelated – if that's the case, why should forest be a predictor for urban trees and pasture for other green spaces?

>>>> They are also correlated. However, it has been shown that that trees/forests have different biophysical climate impacts than grassland or pastures, as we also mention in response to comment 1.5. Thus it seems to make more sense to compare the ΔT (rural forests-urban fabric) with the ΔT (urban trees-urban fabric) and the ΔT (rural pastures-urban fabric) with the ΔT (treeless urban green spaces – urban fabric). But of course trees in urban areas are subject to different conditions and are often composed of different species than outside of urban areas. The same is true for pastures that are not directly comparable with treeless urban green spaces that are mainly urban park areas without trees (i.e. some form of grassland). Even though there are differences between the vegetation in- and outside of cities our results indicate that overall the European pattern of the temperature differences between urban fabric and vegetation in- and outside of cities are similar. This indicates that overall the influence of environmental conditions (i.e. background climate), which are similar inside and outside of the city, have a stronger influence on the ΔT s than the differences in environmental conditions between in- and outside of cities, which may be much more affected by human activity (e.g., irrigation, CO₂ and pollution levels and soil compaction. A more detailed discussion of this point can be found in the introduction (in response to comment 3.1) and in response to comment comment 2.2).

3.36) Figure S2 (now Figure S4): what is the x-axis? It would nice to see if there is a trend with some specific (e.g. climatic) variables

>>>> The x-axis corresponds to the different regions. We did not include x-axis labels, because the figure legend with the different colors allows to identify each box (i.e. each region). We now also included a plot showing average summertime temperature (based on E-OBS) and albedo of urban areas/forested areas. They indicate that there may be a small trend between albedo of urban areas and background temperatures and hardly any trend between albedo of forested areas and background temperatures.

Figure S 19: Scatterplots of background temperatures and albedo values of urban areas (left) and forested areas (right).

3.37) Figure S3 (now Figure S6): it would be useful to see an actual comparison here (i.e. STL vs Lidar).

>>>> We are not entirely sure, what is meant by an actual comparison since both datasets (the Street tree layer and the Lidar data) are shown so that one can compare and see certain differences. Maybe by an actual comparison it is meant to color the differences in a different way. However, we believe that this will not change our main point here, which is that the Street Tree Layer has weaknesses that should be kept in mind.

3.38) Figure S6: write Athens and Bucarest (no Athina and Bucuresti). This should be checked throughout the manuscript.

>>>> We have changed the names of the cities in the whole manuscript. In Table S1 we added English names to several cities, but also kept the previous city names which are unique identifiers for the Copernicus data (i.e. urban atlas and street tree layer).

References

Li, D., Liao, W., Rigden, A. J., Liu, X., Wang, D., Malyshev, S., & Shevliakova, E. (2019). Urban heat island: Aerodynamics or imperviousness?. *Science Advances*, 5(4), eaau4299.

Li, M., Koks, E., Taubenböck, H., & van Vliet, J. (2020). Continental-scale mapping and analysis of 3D building structure. *Remote Sensing of Environment*, 245, 111859.

Rahman, M. A., Stratopoulos, L. M., Moser-Reischl, A., Zölch, T., Häberle, K. H., Rötzer, T., ... & Pauleit, S. (2020). Traits of trees for cooling urban heat islands: A meta-analysis. *Building and Environment*, 170, 106606.

Wang, C., Wang, Z. H., & Yang, J. (2018). Cooling effect of urban trees on the built environment of contiguous United States. *Earth's Future*, 6(8), 1066-1081.

Wang, C., Wang, Z. H., Wang, C., & Myint, S. W. (2019). Environmental cooling provided by urban trees under extreme heat and cold waves in US cities. *Remote sensing of environment*, 227, 28-43.

Winbourne, J. B., Jones, T. S., Garvey, S. M., Harrison, J. L., Wang, L., Li, D., ... & Hutryra, L. R. (2020). Tree Transpiration and Urban Temperatures: Current Understanding, Implications, and Future Research Directions. *BioScience*, 70(7), 576-588.

Zhao, L., Lee, X., Smith, R. B., & Oleson, K. (2014). Strong contributions of local background climate to urban heat islands. *Nature*, 511(7508), 216-219.

References

- ALONSO, L. & RENARD, F. 2020. A New Approach for Understanding Urban Microclimate by Integrating Complementary Predictors at Different Scales in Regression and Machine Learning Models. *Remote Sensing*, 12, 35.
- BEALE, C. M., LENNON, J. J., YEARSLEY, J. M., BREWER, M. J. & ELSTON, D. A. 2010. Regression analysis of spatial data. *Ecology Letters*, 13, 246-264.
- BECHTEL, B., DEMUZERE, M., MILLS, G., ZHAN, W., SISMANIDIS, P., SMALL, C. & VOOGT, J. 2019. SUHI analysis using Local Climate Zones—A comparison of 50 cities. *Urban Climate*, 28, 100451.
- BENALI, A., CARVALHO, A. C., NUNES, J. P., CARVALHAIS, N. & SANTOS, A. 2012. Estimating air surface temperature in Portugal using MODIS LST data. *Remote Sensing of Environment*, 124, 108-121.
- BRONDFIELD, M. N., HUTYRA, L. R., GATELY, C. K., RACITI, S. M. & PETERSON, S. A. 2012. Modeling and validation of on-road CO₂ emissions inventories at the urban regional scale. *Environmental Pollution*, 170, 113-123.
- BURAKOWSKI, E., TAWFIK, A., OUIMETTE, A., LEPINE, L., NOVICK, K., OLLINGER, S., ZARZYCKI, C. & BONAN, G. 2018. The role of surface roughness, albedo, and Bowen ratio on ecosystem energy balance in the Eastern United States. *Agricultural and Forest Meteorology*, 249, 367-376.
- CARLSON, T. N. & RIPLEY, D. A. 1997. On the relation between NDVI, fractional vegetation cover, and leaf area index. *Remote Sensing of Environment*, 62, 241-252.
- CHAKRABORTY, T., HSU, A., MANYA, D. & SHERIFF, G. 2020. A spatially explicit surface urban heat island database for the United States: Characterization, uncertainties, and possible applications. *ISPRS Journal of Photogrammetry and Remote Sensing*, 168, 74-88.
- CHEN, X. P., ZHOU, Z. X., TENG, M. J., WANG, P. C. & ZHOU, L. 2015. ACCUMULATION OF THREE DIFFERENT SIZES OF PARTICULATE MATTER ON PLANT LEAF SURFACES: EFFECT ON LEAF TRAITS. *Archives of Biological Sciences*, 67, 1257-1267.
- CHRISTIDIS, N. & STOTT, P. A. 2021. The influence of anthropogenic climate change on wet and dry summers in Europe. *Science Bulletin*.
- CHRYSOULAKIS, N., GRIMMOND, S., FEIGENWINTER, C., LINDBERG, F., GASTELLU-ETCHEGORRY, J.-P., MARCONCINI, M., MITRAKA, Z., STAGAKIS, S., CRAWFORD, B., OLOFSON, F., LANDIER, L., MORRISON, W. & PARLOW, E. 2018. Urban energy exchanges monitoring from space. *Scientific Reports*, 8, 11498.
- COPERNICUS 2016. Mapping Guide v4.7 for a European Urban Atlas.
- CORNES, R. C., VAN DER SCHRIER, G., VAN DEN BESSELAAR, E. J. M. & JONES, P. D. 2018. An Ensemble Version of the E-OBS Temperature and Precipitation Data Sets. *Journal of Geophysical Research: Atmospheres*, 123, 9391-9409.
- COUTTS, A. M., WHITE, E. C., TAPPER, N. J., BERINGER, J. & LIVESLEY, S. J. 2016. Temperature and human thermal comfort effects of street trees across three contrasting street canyon environments. *Theoretical and Applied Climatology*, 124, 55-68.
- CZAJA, M., KOŁTON, A. & MURAS, P. 2020. The Complex Issue of Urban Trees—Stress Factor Accumulation and Ecological Service Possibilities. *Forests*, 11, 932.
- DAVIN, E. L., SENEVIRATNE, S. I., CIAIS, P., OLIOSO, A. & WANG, T. 2014. Preferential cooling of hot extremes from cropland albedo management. *Proceedings of the National Academy of Sciences of the United States of America*, 111, 9757-9761.
- DE KAUWE, M. G., MEDLYN, B. E., PITMAN, A. J., DRAKE, J. E., UKKOLA, A., GRIEBEL, A., PENDALL, E., PROBER, S. & RODERICK, M. 2019. Examining the evidence for decoupling between photosynthesis and transpiration during heat extremes. *Biogeosciences*, 16, 903-916.

- DEBBAGE, N. & SHEPHERD, J. M. 2015. The urban heat island effect and city contiguity. *Computers, Environment and Urban Systems*, 54, 181-194.
- DECINA, S. M., TEMPLER, P. H., HUTYRA, L. R., GATELY, C. K. & RAO, P. 2017. Variability, drivers, and effects of atmospheric nitrogen inputs across an urban area: Emerging patterns among human activities, the atmosphere, and soils. *Science of The Total Environment*, 609, 1524-1534.
- DENISSEN, J. M. C., TEULING, A. J., REICHSTEIN, M. & ORTH, R. 2020. Critical Soil Moisture Derived From Satellite Observations Over Europe. *Journal of Geophysical Research: Atmospheres*, 125, e2019JD031672.
- DUVEILLER, G., HOOKER, J. & CESCATTI, A. 2018. The mark of vegetation change on Earth's surface energy balance. *Nature Communications*, 9, 679.
- GAO, K. & SANTAMOURIS, M. 2019. The use of water irrigation to mitigate ambient overheating in the built environment: Recent progress. *Building and Environment*, 164, 8.
- GILLNER, S., VOGT, J., THARANG, A., DETTMANN, S. & ROLOFF, A. 2015. Role of street trees in mitigating effects of heat and drought at highly sealed urban sites. *Landscape and Urban Planning*, 143, 33-42.
- GIOMETTO, M. G., CHRISTEN, A., EGLI, P. E., SCHMID, M. F., TOOKE, R. T., COOPS, N. C. & PARLANGE, M. B. 2017. Effects of trees on mean wind, turbulence and momentum exchange within and above a real urban environment. *Advances in Water Resources*, 106, 154-168.
- GOOD, E. J. 2016. An in situ-based analysis of the relationship between land surface "skin" and screen-level air temperatures. *Journal of Geophysical Research: Atmospheres*, 121, 8801-8819.
- HARDIN, P. J. & JENSEN, R. R. 2007. The effect of urban leaf area on summertime urban surface kinetic temperatures: A Terre Haute case study. *Urban Forestry & Urban Greening*, 6, 63-72.
- HARI, V., RAKOVEC, O., MARKONIS, Y., HANEL, M. & KUMAR, R. 2020. Increased future occurrences of the exceptional 2018–2019 Central European drought under global warming. *Scientific Reports*, 10, 12207.
- HARRISON, J. L., REINMANN, A. B., MALONEY, A. S., PHILLIPS, N., JUICE, S. M., WEBSTER, A. J. & TEMPLER, P. H. 2020. Transpiration of Dominant Tree Species Varies in Response to Projected Changes in Climate: Implications for Composition and Water Balance of Temperate Forest Ecosystems. *Ecosystems*, 23, 1598-1613.
- HO, H. C., KNUDBY, A., XU, Y., HODUL, M. & AMINIPOURI, M. 2016. A comparison of urban heat islands mapped using skin temperature, air temperature, and apparent temperature (Humidex), for the greater Vancouver area. *Science of The Total Environment*, 544, 929-938.
- HOOKE, J., DUVEILLER, G. & CESCATTI, A. 2018. A global dataset of air temperature derived from satellite remote sensing and weather stations. *Scientific Data*, 5, 180246.
- HU, Y., HOU, M., JIA, G., ZHAO, C., ZHEN, X. & XU, Y. 2019. Comparison of surface and canopy urban heat islands within megacities of eastern China. *ISPRS Journal of Photogrammetry and Remote Sensing*, 156, 160-168.
- IIO, A., HIKOSAKA, K., ANTEN, N. P. R., NAKAGAWA, Y. & ITO, A. 2014. Global dependence of field-observed leaf area index in woody species on climate: a systematic review. *Global Ecology and Biogeography*, 23, 274-285.
- IMHOFF, M. L., ZHANG, P., WOLFE, R. E. & BOUNOUA, L. 2010. Remote sensing of the urban heat island effect across biomes in the continental USA. *Remote Sensing of Environment*.
- JIM, C. Y. 2019. Soil volume restrictions and urban soil design for trees in confined planting sites. *Journal of Landscape Architecture*, 14, 84-91.
- KLOOG, I., CHUDNOVSKY, A., KOUTRAKIS, P. & SCHWARTZ, J. 2012. Temporal and spatial assessments of minimum air temperature using satellite surface temperature measurements in Massachusetts, USA. *Science of The Total Environment*, 432, 85-92.

- KRAYENHOFF, E. S. & VOOGT, J. A. 2016. Daytime Thermal Anisotropy of Urban Neighbourhoods: Morphological Causation. *Remote Sensing*, 8, 108.
- KROEGER, T., MCDONALD, R. I., BOUCHER, T., ZHANG, P. & WANG, L. 2018. Where the people are: Current trends and future potential targeted investments in urban trees for PM10 and temperature mitigation in 27 U.S. Cities. *Landscape and Urban Planning*, 177, 227-240.
- LEWIS, P. & BARNESLEY, M. 1994. Influence of the sky radiance distribution on various formulations of the Earth surface albedo. *Proc. Conf. Phys. Meas. Sign. Remote Sens.*
- LI, D., LIAO, W., RIGDEN, A. J., LIU, X., WANG, D., MALYSHEV, S. & SHEVLIAKOVA, E. 2019. Urban heat island: Aerodynamics or imperviousness? *Science Advances*, 5, eaau4299.
- LI, M., KOKS, E., TAUBENBÖCK, H. & VAN VLIET, J. 2020. Continental-scale mapping and analysis of 3D building structure. *Remote Sensing of Environment*, 245, 111859.
- LI, Y., ZHAO, M., MOTESHARREI, S., MU, Q., KALNAY, E. & LI, S. 2015. Local cooling and warming effects of forests based on satellite observations. *Nature Communications*, 6.
- LOGAN, T. M., ZAITCHIK, B., GUIKEMA, S. & NISBET, A. 2020. Night and day: The influence and relative importance of urban characteristics on remotely sensed land surface temperature. *Remote Sensing of Environment*, 247, 111861.
- MANOLI, G., FATICHI, S., BOU-ZEID, E. & KATUL, G. G. 2020a. Seasonal hysteresis of surface urban heat islands. *Proceedings of the National Academy of Sciences*, 117, 7082.
- MANOLI, G., FATICHI, S., SCHLÄPFER, M., YU, K., CROWTHER, T., MEILI, N., BURLANDO, P., KATUL, G. & BOU-ZEID, E. 2020b. *Reply to Martilli et al. (2020): Summer average urban-rural surface temperature differences do not indicate the need for urban heat reduction.*
- MARTILLI, A., ROTH, M., CHOW, W., DEMUZERE, M., LIPSON, M., KRAYENHOFF, E., SAILOR, D., NAZARIAN, N., VOOGT, J., WOUTERS, H., MIDDEL, A., STEWART, I., BECHTEL, B., CHRISTEN, A. & HART, M. 2020. *Summer average urban-rural surface temperature differences do not indicate the need for urban heat reduction.*
- MCADAM, S. A. M. & BRODRIBB, T. J. 2015. The Evolution of Mechanisms Driving the Stomatal Response to Vapor Pressure Deficit. *Plant Physiology*, 167, 833-843.
- MEILI, N., MANOLI, G., BURLANDO, P., CARMELIET, J., CHOW, W. T. L., COUTTS, A. M., ROTH, M., VELASCO, E., VIVONI, E. R. & FATICHI, S. 2021. Tree effects on urban microclimate: Diurnal, seasonal, and climatic temperature differences explained by separating radiation, evapotranspiration, and roughness effects. *Urban Forestry & Urban Greening*, 58, 126970.
- MEINEKE, E., YOUNGSTEADT, E., DUNN, R. R. & FRANK, S. D. 2016. Urban warming reduces aboveground carbon storage. *Proceedings of the Royal Society B: Biological Sciences*, 283, 20161574.
- MELAAS, E. K., WANG, J. A., MILLER, D. L. & FRIEDL, M. A. 2016. Interactions between urban vegetation and surface urban heat islands: a case study in the Boston metropolitan region. *Environmental Research Letters*, 11, 054020.
- MILDREXLER, D. J., ZHAO, M. & RUNNING, S. W. 2011. A global comparison between station air temperatures and MODIS land surface temperatures reveals the cooling role of forests. *Journal of Geophysical Research: Biogeosciences*, 116.
- MUSSETTI, G., BRUNNER, D., HENNE, S., ALLEGRI, J., KRAYENHOFF, E. S., SCHUBERT, S., FEIGENWINTER, C., VOGT, R., WICKI, A. & CARMELIET, J. 2020. COSMO-BEP-Tree v1.0: a coupled urban climate model with explicit representation of street trees. *Geoscientific Model Development*, 13, 1685-1710.
- MUTIIBWA, D., STRACHAN, S. & ALBRIGHT, T. 2015. Land Surface Temperature and Surface Air Temperature in Complex Terrain. *Ieee Journal of Selected Topics in Applied Earth Observations and Remote Sensing*, 8, 4762-4774.
- NOVICK, K. A. & KATUL, G. G. 2020. The Duality of Reforestation Impacts on Surface and Air Temperature. *Journal of Geophysical Research: Biogeosciences*, 125, e2019JG005543.

- PARASTATIDIS, D., MITRAKA, Z., CHRYSOULAKIS, N. & ABRAMS, M. 2017. Online Global Land Surface Temperature Estimation from Landsat. *Remote Sensing*, 9, 16.
- PATAKI, D. E., MCCARTHY, H. R., LITVAK, E. & PINCETL, S. 2011. Transpiration of urban forests in the Los Angeles metropolitan area. *Ecological Applications*, 21, 661-677.
- PAULEIT, S., JONES, N., GARCIA-MARTIN, G., GARCIA-VALDECANTOS, J. L., RIVIÈRE, L. M., VIDAL-BEAUDET, L., BODSON, M. & RANDRUP, T. B. 2002. Tree establishment practice in towns and cities – Results from a European survey. *Urban Forestry & Urban Greening*, 1, 83-96.
- PERKINS, S. E. 2015. A review on the scientific understanding of heatwaves—Their measurement, driving mechanisms, and changes at the global scale. *Atmospheric Research*, 164-165, 242-267.
- RAHMAN, M. A., MOSER, A., ROTZER, T. & PAULEIT, S. 2019. Comparing the transpirational and shading effects of two contrasting urban tree species. *Urban Ecosystems*, 22, 683-697.
- RAHMAN, M. A., STRATOPOULOS, L. M. F., MOSER-REISCHL, A., ZÖLCH, T., HÄBERLE, K.-H., RÖTZER, T., PRETZSCH, H. & PAULEIT, S. 2020. Traits of trees for cooling urban heat islands: A meta-analysis. *Building and Environment*, 170, 106606.
- REYES-PAECKE, S., GIRONAS, J., MELO, O., VICUNA, S. & HERRERA, J. 2019. Irrigation of green spaces and residential gardens in a Mediterranean metropolis: Gaps and opportunities for climate change adaptation. *Landscape and Urban Planning*, 182, 34-43.
- ROMAN, D. T., NOVICK, K. A., BRZOSTEK, E. R., DRAGONI, D., RAHMAN, F. & PHILLIPS, R. P. 2015. The role of isohydric and anisohydric species in determining ecosystem-scale response to severe drought. *Oecologia*, 179, 641-654.
- ROTENBERG, E. & YAKIR, D. 2010. Contribution of Semi-Arid Forests to the Climate System. *Science*, 327, 451-454.
- SCHENK, H. J. & JACKSON, R. B. 2002. THE GLOBAL BIOGEOGRAPHY OF ROOTS. *Ecological Monographs*, 72, 311-328.
- SCHWARZ, N. & MANCEUR, A. M. 2015. Analyzing the Influence of Urban Forms on Surface Urban Heat Islands in Europe. *Journal of Urban Planning and Development*.
- SERRA, C., LANA, X., MARTÍNEZ, M. D., ROCA, J., ARELLANO, B., BIERE, R., MOIX, M. & BURGUEÑO, A. 2020. Air temperature in Barcelona metropolitan region from MODIS satellite and GIS data. *Theoretical and Applied Climatology*, 139, 473-492.
- SHASHUA-BAR, L., PEARLMUTTER, D. & ERELL, E. 2009. The cooling efficiency of urban landscape strategies in a hot dry climate. *Landscape and Urban Planning*, 92, 179-186.
- SMITHERS, R. J., DOICK, K. J., BURTON, A., SIBILLE, R., STEINBACH, D., HARRIS, R., GROVES, L. & Blicharska, M. 2018. Comparing the relative abilities of tree species to cool the urban environment. *Urban Ecosystems*, 21, 851-862.
- SU, Y., LIU, L., LIAO, J., WU, J., CIAIS, P., LIAO, J., HE, X., LIU, X., CHEN, X., YUAN, W., ZHOU, G. & LAFORTEZZA, R. 2020. Phenology acts as a primary control of urban vegetation cooling and warming: A synthetic analysis of global site observations. *Agricultural and Forest Meteorology*, 280, 107765.
- TESKEY, R., WERTIN, T., BAUWERAERTS, I., AMEYE, M., MCGUIRE, M. A. & STEPPE, K. 2015. Responses of tree species to heat waves and extreme heat events. *Plant, Cell & Environment*, 38, 1699-1712.
- TEULING, A. J., SENEVIRATNE, S. I., STOCKLI, R., REICHSTEIN, M., MOORS, E., CIAIS, P., LUYSSAERT, S., VAN DEN HURK, B., AMMANN, C., BERNHOFER, C., DELLWIK, E., GIANELLE, D., GIELEN, B., GRUNWALD, T., KLUMPP, K., MONTAGNANI, L., MOUREAUX, C., SOTTOCORNOLA, M. & WOHLFAHRT, G. 2010. Contrasting response of European forest and grassland energy exchange to heatwaves. *Nature Geoscience*, 3, 722-727.
- TSIROU, I. X. 2010. Assessment and energy implications of street air temperature cooling by shade trees in Athens (Greece) under extremely hot weather conditions. *Renewable Energy*, 35, 1866-1869.

- WANG, C. H., WANG, Z. H., WANG, C. Y. & MYINT, S. W. 2019a. Environmental cooling provided by urban trees under extreme heat and cold waves in US cities. *Remote Sensing of Environment*, 227, 28-43.
- WANG, C. H., WANG, Z. H. & YANG, J. C. 2018. Cooling Effect of Urban Trees on the Built Environment of Contiguous United States. *Earths Future*, 6, 1066-1081.
- WANG, J. H., CAI, X. M. & VALOCCHI, A. 2015. Spatial Evolutionary Algorithm for Large-Scale Groundwater Management. In: SUN, H., YANG, C. Y., LIN, C. W., PAN, J. S., SNASEL, V. & ABRAHAM, A. (eds.) *Genetic and Evolutionary Computing*. Berlin: Springer-Verlag Berlin.
- WANG, L., HUANG, M. & LI, D. 2020. Where Are White Roofs More Effective in Cooling the Surface? *Geophysical Research Letters*, 47, e2020GL087853.
- WANG, P., LI, D., LIAO, W. L., RIGDEN, A. & WANG, W. 2019b. Contrasting Evaporative Responses of Ecosystems to Heatwaves Traced to the Opposing Roles of Vapor Pressure Deficit and Surface Resistance. *Water Resources Research*, 55, 4550-4563.
- WINBOURNE, J. B., JONES, T. S., GARVEY, S. M., HARRISON, J. L., WANG, L., LI, D., TEMPLER, P. H. & HUTYRA, L. R. 2020. Tree Transpiration and Urban Temperatures: Current Understanding, Implications, and Future Research Directions. *BioScience*, 70, 576-588.
- WU, Z., YAO, L. & REN, Y. 2020. Characterizing the spatial heterogeneity and controlling factors of land surface temperature clusters: A case study in Beijing. *Building and Environment*, 169, 106598.
- YOSEF, G., WALKO, R., AVISAR, R., TATARINOV, F., ROTENBERG, E. & YAKIR, D. 2018. Large-scale semi-arid afforestation can enhance precipitation and carbon sequestration potential. *Scientific Reports*, 8, 996.
- ZHANG, P., BOUNOUA, L., IMHOFF, M. L., WOLFE, R. E. & THOME, K. 2014. Comparison of MODIS Land Surface Temperature and Air Temperature over the Continental USA Meteorological Stations. *Canadian Journal of Remote Sensing*, 40, 110-122.
- ZHAO, L., LEE, X., SMITH, R. B. & OLESON, K. 2014. Strong contributions of local background climate to urban heat islands. *Nature*, 511, 216-219.
- ZHOU, W., HUANG, G. & CADENASSO, M. L. 2011. Does spatial configuration matter? Understanding the effects of land cover pattern on land surface temperature in urban landscapes. *Landscape and Urban Planning*, 102, 54-63.
- ZIPPER, S. C., SCHATZ, J., KUCHARIK, C. J. & LOHEIDE II, S. P. 2017. Urban heat island-induced increases in evapotranspirative demand. *Geophysical Research Letters*, 44, 873-881.

REVIEWER COMMENTS

Reviewer #1 (Remarks to the Author):

Summary:

In the revised manuscript, now appropriately retitled "The Role of Urban Trees in Reducing Land Surface Temperatures in European Cities", the authors have made substantial improvements, both in the methodology and the discussion of the limitations of the study. Now that I have had more clarification on these aspects though, I am a bit concerned about the robustness of the results given the number of uncertainties involved.

Major comments:

1. It is true that urban vegetation reduces temperature (both surface and air) and that trees would have a higher cooling potential than other green spaces due to their generally higher evapotranspiration. The authors have shown this is to be the case across most cities, which supports the results of countless studies before (with the present study doing it for multiple cities at once). However, given the statistical nature of the models used, the strength of this cooling potential is quite dependent on how the factors are calculated. The authors show bar plots comparing the impact of emissivity from different sources on LST for a subset of cities. However, this shows the impact on absolute LST, while the authors mostly focus on LST differentials, where the impact of emissivity would be a lot more apparent. Similarly, they compare the black sky and white sky albedo using bar plots. Though the exact values are not given, from Fig. S14 it seems like the differences can be as high as 0.03. This may seem small but is non-negligible when you realize that it is scaled by the incoming shortwave radiation during daytime. As such, one would wonder how the overall results would change if these alternate ways of calculating the factors (different emissivity assumptions for deriving LST and actual diffuse fraction to calculate total albedo) were used. This methodological issue makes it hard to determine how robust the results are.
2. The objective of the study is still focused on the cooling potential of green spaces with a focus on urban heat, which is not the same as LST, which is studied here. The added discussion on air and surface temperature adds some perspective to these issues. However, note that a big part of the cooling potential of trees is the shading effect, which cannot be extracted using satellite-observed thermal data. What the study compares is the canopy-top temperature of trees versus the treeless green spaces. As such, it is difficult to say how useful this information will be to policymakers or if it should even be used for policymaking without significant qualifiers.

Minor comments:

1. The choice of using E-OBS as the background temperature is difficult to justify. Since the analysis is done using LST, using air temperature for the background climate adds a new confounding factor. The coupling between air and surface temperature is dependent on the background climate, particularly the density of vegetation. As such, this is not a good methodological choice. Instead, it might be better to use reanalysis that also provides LST for each grid. Most reanalysis products do not explicitly include urban land units. However, it is important to first evaluate whether the reanalysis products have any systematic bias (compared to the satellite-derived LST) before its inclusion.

Reviewer #2 (Remarks to the Author):

The authors have revised the manuscript accordingly. I am also satisfied with their response to my comments. I have no further questions.

Reviewer #3 (Remarks to the Author):

The authors carefully addressed all the comments provided by me and the other reviewers and, in my opinion, the quality of the manuscript has markedly improved. I only have a few additional comments.

Abstract: I like the changes but I would highlight the fact that the results here can inform cooling strategies/policies on the best type of "green spaces" as the last sentence (lines 24-27) is a bit vague. In other words, the key message is that trees are more "efficient" than treeless green spaces but such efficiency varies across climatic regions and average/extreme weather conditions – this is the key finding with policy relevance that should be highlighted.

Actually, a recent study has come to similar conclusions (Paschalis et al. 2021) so a discussion/comparison would be useful (e.g. in the introduction or discussion section). In this context, I agree with the comment of reviewer 1 (see comment 1.5 in the rebuttal) that the focus on trees vs no-trees is somewhat limited compared to NDVI or LAI-based approaches but might be more "useful" for policy decisions (even if, as discussed by the authors, the definition of "trees" is also limited, as a line of street trees is certainly different from a dense urban forest). Given the potential interest of the topic for a wide audience, a few clarifications along these lines might be beneficial.

Introduction: I appreciate the additions made by the authors on the physical mechanisms leading to cooling by trees. However, I would suggest to partly restructure this section to improve the logical flow of thoughts. For example, the fact that vegetation "can either increase or decrease temperature" is repeated in Line 68 and 73, the effect of water stress is mentioned in Line 66 and 78, etc.

In general, the writing/language still needs some improvements (see below for a few suggestions).

Line 37: consider rephrasing as "potential cooling effect of trees in ..."

Line 42: consider rephrasing as "as demonstrated by several studies"

Line 48: remove "e.g." or rephrase ("e.g. see ref. 7"). Check this throughout the manuscript.

Line 52: "mainly on ..."

Lines 55-56: the effect of temperature on stomatal conductance is true for vegetation in general, not just trees – please clarify

Line 63: not sure "ET" has been defined before

Line 99: "LST observations"

Line 117: "regions of France, ..."

Line 239: "treeless urban green spaces"?

Lines 252-255: what is the reasoning behind this statement? Clarify

Line 298: I think "surrounding", rather than "neighbourhood", is more appropriate here

Line 300: thanks for clarifying this – however, the sentence is still unclear to me. The fact that urban fabric has little or no ET is indeed expected but it is not a "minor effect" on LST differences – such differences are high precisely because ET is low over urban surfaces. I suggest rephrasing this.

Lines 320-324: see Paschalis et al. (2021) for a discussion on urban-rural differences in LAI and SIF.

Line 335: I think MODIS is mentioned here for the first time – either explain upfront or write something like "the MODIS ET product used here..."

Line 388: this sentence sounds unclear to me

Lines 400-401: unclear how LCZs are related to the work here – is this a suggestion for improvement of the method? Is it an alternative? Please elaborate (...ok, now I see that the sentence is different in the rebuttal, please correct the main text).

Line 427: socio-economic factors? What does this mean?

Line 429: this statement is "out of the blue". I suggest rephrasing as "In conclusion, we presented"

Line 431: UHI data are generally obtained from LST and LULC data which are then aggregated at the city-scale – so not sure I agree with this sentence.

Figure 2: are the results in panel d also illustrated in panel c? If yes, I would remove panel d or simplify/clarify. Also, what do colours in panel d (boxplots) indicate?

Figure 3: clarify in the legend that urban green spaces are treeless

Figure S1: what is the legend for the colours in panel a? The colorbar in panel b seems to show a different temperature range

Figure S3: I would use the same temperature limits/colorbar for the top and bottom panels

Figure S4: for consistency, urban fabric should be indicated by UF

Figures S15-S20 are not discussed in the text – I suggest adding a supplementary section motivating/explaining these analyses. Actually, all these supplementary figures/results should be mentioned also in the main text (not sure whether this is a journal requirement but it is certainly useful for the reader).

References

Paschalis, A., Chakraborty, T. C., Fatichi, S., Meili, N., & Manoli, G. (2021). Urban forests as main regulator of the evaporative cooling effect in cities. *AGU Advances*, 2(2), e2020AV000303.

We thank all reviewers for their time and effort and their very helpful and constructive comments. In the following, we list the reviewers' comments in black and our reply in blue.

Reviewer #1 (Remarks to the Author):

Summary:

In the revised manuscript, now appropriately retitled "The Role of Urban Trees in Reducing Land Surface Temperatures in European Cities", the authors have made substantial improvements, both in the methodology and the discussion of the limitations of the study. Now that I have had more clarification on these aspects though, I am a bit concerned about the robustness of the results given the number of uncertainties involved.

Major comments:

1.1 It is true that urban vegetation reduces temperature (both surface and air) and that trees would have a higher cooling potential than other green spaces due to their generally higher evapotranspiration. The authors have shown this is to be the case across most cities, which supports the results of countless studies before (with the present study doing it for multiple cities at once). However, given the statistical nature of the models used, the strength of this cooling potential is quite dependent on how the factors are calculated. The authors show bar plots comparing the impact of emissivity from different sources on LST for a subset of cities. However, this shows the impact on absolute LST, while the authors mostly focus on LST differentials, where the impact of emissivity would be a lot more apparent.

>>>> We indeed show plots in Figure S 8 for comparing the impact of different emissivity sources on LST. However, there may be a misunderstanding here. We show the influence of different emissivity sources on the LST difference between urban trees and urban fabric and not the influence of different emissivities on absolute LST. To clarify this, we updated the y-axis labels in Figure S 8 so that they explicitly refer to the difference between urban trees and urban fabric. This had previously only been mentioned in the figure caption. The difference in LST (between urban trees and urban fabric) for different emissivity sources can be up to 1.5 K, but usually the LSTs for the different emissivity sources are very close together. To make this finding more explicit, we include Figure S 21, which shows the scatterplots/correlations of LST differences (urban trees minus urban fabric) for different emissivity sources and different cities. Overall, the tested sensitivities indicate that the NDVI based LST differences could be slightly higher than when using emissivities based on ASTER or MODIS. We included the following statement into the manuscript:

"The LST differences between urban trees and urban fabric are robust to different choices of emissivity sources (Figure S 8, Figure S 21). However, it should be noted that NDVI-based emissivities may generally produce slightly higher LST differences than Aster or Modis-based emissivities (Figure S 8)."

Figure S 8: Influence of different emissivities chosen for calculating the Landsat LST product. The temperature differences between urban trees and continuous urban fabric are very similar for all emissivities. The NDVI-based emissivity seems to lead to a slightly higher temperature difference than the other emissivities.

Figure S 21: LST differences between urban trees and urban fabric based on NDVI emissivity plotted against LST differences between urban trees and urban fabric based on MODIS and ASTER emissivity.

1.2 Similarly, they compare the black sky and white sky albedo using bar plots. Though the exact values are not given, from Fig. S14 it seems like the differences can be as high as 0.03. This may seem small but is non-negligible when you realize that it is scaled by the incoming shortwave radiation during daytime. As such, one would wonder how the overall results would change if these alternate ways of calculating the factors (different emissivity assumptions for deriving LST and actual diffuse fraction to calculate total albedo) were used. This methodological issue makes it hard to determine how robust the results are.

>>>> We think it is important to clarify here that we do not in any way use the albedo to calculate or correct the LST data that we are using. We use albedo and evapotranspiration to better understand how the inter-city LST differences (i.e. regional patterns) that we observe may be explained by those two variables (by correlating the LST differences with albedo and ET differences). The albedo uncertainty will therefore not influence any of the results presented in the main part of the manuscript. The only thing that might change is that the albedo could have a stronger correlation with the regional LST patterns than our data currently shows and hence may be a more important factor. Based on previous reviewer comments we had looked at the differences in black-sky and white-sky albedo and created Figure S 14 to show that the regional patterns are the same for black- and white-sky albedo. Blue-sky albedo is therefore likely to follow the same patterns. This would only change if there was a strong systematic variation in regional patterns of the ratios between diffuse and direct radiation. Even though we believe that these variations are rather small for clear-sky conditions (for which LST data is available) we tested the sensitivity of assuming different ratios of direct and diffuse radiation and hence different weights of black- and white-sky albedo. The results show that there is no substantial correlation between the albedo and the LST patterns that we observe (Figure S 23), which is indeed in line with the results of

other studies that show that the effect of albedo (and roughness) may be small in comparison to the effect of ET (Manoli et al., 2019, Li et al., 2019). But of course, as we also point out in the discussion, the effect of albedo may be particularly relevant in southern European regions, where applying concepts of energy redistribution would suggest that low latent heat fluxes could increase the relevance of albedo (Wang et al., 2020). To further illustrate the sensitivity of the albedo results concerning the fraction of white- and black-sky albedo we now show figures on the correlation and the differences between white- and black-sky albedo (Figure S 22). In addition, we show correlations of the LST difference of urban fabric and urban trees with different combinations of black- and white-sky (Figure S 23).

Figure S 22: Correlation and differences between black- and white-sky albedo of European cities. a) Correlation between black- and white-sky albedo of urban fabric. b) Differences between black- and white-sky albedo of urban fabric. c) Correlation between black- and white-sky albedo of forests. d) Differences between black- and white-sky albedo of urban fabric.

Figure S 23: Correlations between LST differences (urban trees minus urban fabric) and albedo differences (forests minus urban fabric). The albedo differences are calculated as $\Delta \alpha_{F-UF} = \alpha_F - \alpha_{UF}$ with $\alpha_F = \omega_F \alpha_{F,WSA} + (1 - \omega_F) \alpha_{UF,BSA}$ and $\alpha_{UF} = \omega_{UF} \alpha_{UF,WSA} + (1 - \omega_{UF}) \alpha_{UF,BSA}$, where ω can be understood as the weight given to white- and black-sky albedo or as the ratio of the surface downward diffuse shortwave radiation to the surface downward total shortwave radiation (Wang et al., 2015). The weights (i.e. ratios) ω_F and ω_{UF} were varied in several ways between 0 and 1 to test how these choices influence the correlation between LST and albedo. Regions are numbered in the following way: Mediterranean (1), Iberian Peninsula (2), Turkey (3), British Isles (4), France (5), Alps/Mid-Europe

(6), Eastern Europe (7) and Scandinavia (8) a) Equal weight ($\omega_F = 0.5, \omega_{UF} = 0.5$) given to WSA and BSA for all cities and both land cover types (i.e. urban fabric and forest) b) BSA for all cities and land-cover types ($\omega_F = 0, \omega_{UF} = 0$). c) WSA for all cities and land-cover types ($\omega_F = 1, \omega_{UF} = 1$). d) WSA to calculate forest albedo and BSA to calculate urban fabric albedo ($\omega_F = 1, \omega_{UF} = 0$). e) BSA to calculate forest albedo and WSA to calculate urban fabric albedo ($\omega_F = 0, \omega_{UF} = 1$). f) BSA/WSA to calculate forest albedo and WSA to calculate urban fabric albedo in the regions 1,2 and 3 ($\omega_F = 0.5, \omega_{UF} = 1$), BSA to calculate urban fabric albedo and BSA/WSA to calculate forest albedo in the regions 4, 5, 6, 7 and 8 ($\omega_F = 0.5, \omega_{UF} = 0$). g) Forest albedo (equally weighed WSA/BSA) minus urban fabric WSA in regions 4,5,6 and 7 and urban fabric BSA in regions 1,2,3 and 8. h) Forest albedo (equally weighed WSA/BSA) minus urban fabric WSA in regions 4,5,6, 7 and 8 and urban fabric BSA in regions 1,2 and 3. j) Forest albedo (equally weighed WSA/BSA) minus urban fabric WSA in regions 8 and urban fabric BSA in regions 1,2, 3, 4, 5, 6 and 7.

2. The objective of the study is still focused on the cooling potential of green spaces with a focus on urban heat, which is not the same as LST, which is studied here. The added discussion on air and surface temperature adds some perspective to these issues. However, note that a big part of the cooling

potential of trees is the shading effect, which cannot be extracted using satellite-observed thermal data. What the study compares is the canopy-top temperature of trees versus the treeless green spaces. As such, it is difficult to say how useful this information will be to policymakers or if it should even be used for policymaking without significant qualifiers.

>>>> We also agree that the shading effect is very important and that this is of course a very important advantage of trees in comparison to treeless green spaces. To better highlight the relevance of shading (in addition to previous mentioning of this effect in the manuscript) we adapted the Discussion section:

“LSTs observed for different vegetation types in different regions can be largely explained by different ET levels, but LST differences do not reflect the radiative cooling (shading effect) provided by trees underneath their canopies. The shading effect of trees may follow different regional patterns than the ET related reduction of LST. For example, shading of trees can be particularly relevant in Mediterranean regions with high amounts of incoming solar radiation. ...”

The strengths of using LST data to study potential effects of vegetation is that the data is consistently available on high resolution over the whole earth. Yet, we of course agree that urban heat and LST are not the same. We think that there are quite clear definitions for the urban heat island effect, but actually not for urban heat, which nevertheless should be the preferred term here (Martilli et al., 2020). While the urban heat island effect can be defined, e.g., based on the height where it is measured (e.g., subsurface UHI, surface UHI, canopy UHI and boundary layer UHI (Oke et al., 2017)), the term urban heat is less clearly defined. A possible definition/description could be the heat integrated over all vertical layers of an area defined as urban. LST would be one possible indicator for urban heat, since it covers one vertical layer and is under many circumstances correlated to temperatures in other vertical layers. However, LST has of course certain limitations in describing overall urban heat and also urban heat where it may have its most adverse effects (as is also mentioned in the discussion on air temperature and LST). Thus, we add another qualifier to make clear that we do not think that our results should be understood, nor be used exclusively in policy- and decision-making, as an indicator for the overall cooling benefits of different vegetation types under different climatic conditions.

“... This highlights once again that our results should not be interpreted as indicating the overall cooling benefits of different vegetation types in different regions. They should be interpreted in combination with results obtained in studies with a different methodological and thematic focus (e.g. based on meteorological in-situ observation and climate modelling experiments focusing, e.g., on air temperature or human thermal comfort). Each approach may have its limitations in terms of spatial coverage, temporal resolution and degree of uncertainty. But looking at results from each of these approaches together can be very relevant when supporting policy- and decision-making.”

In addition, we modified the last sentence of the abstract (see also response to reviewer 3, comment 1), which was the only sentence that had been directly referring to policy and decision-making. While reviewer 3 is suggesting to emphasize the policy implications in this sentence, we now tried to find a compromise. We now point out that our main findings and messages are not per se simplifying decision-making. Instead they show the complex dependencies that policy- and decision-makers should be aware of:

“By revealing continental-scale patterns in the effect of trees and treeless green spaces on urban LST our results highlight the importance of considering and further investigating the climate-dependent effectiveness of heat mitigation measures in cities.”

Minor comments:

1. The choice of using E-OBS as the background temperature is difficult to justify. Since the analysis is done using LST, using air temperature for the background climate adds a new confounding factor. The coupling between air and surface temperature is dependent on the background climate, particularly the density of vegetation. As such, this is not a good methodological choice. Instead, it might be better to use reanalysis that also provides LST for each grid. Most reanalysis products do not explicitly include urban land units. However, it is important to first evaluate whether the reanalysis products have any systematic bias (compared to the satellite-derived LST) before its inclusion.

>>>> We used E-OBS data in response to previous reviewer comments that rightfully pointed out that spatially averaging LST observations (the previous approach) could be an issue since each city may have a different fraction of vegetation and since a specific LST observation (LANDSAT/ASTER scene) can sometimes only cover a fraction of the city and its surrounding (with either more or less vegetation). E-OBS data is in contrast largely independent of the underlying land-cover since it can be roughly understood as an interpolation of station data from the ECA&D (European Climate Assessment & Dataset) initiative. The stations are usually located on standardized plots and to a large degree independent of the land-cover in their surroundings. Thus, we think that E-OBS is a very good choice. We don't see E-OBS and the use of air temperature as a confounding factor, because the air temperature is not included as a predictor variable when calibrating the Generalized Additive Models. The E-OBS temperatures are used to distinguish hot days from less hot days. To make this distinction LST from reanalysis products may also be suitable, but there is no apparent advantage to us. It might even be an issue that certain areas are missing as has been pointed out in the reviewer comment.

Reviewer #2 (Remarks to the Author):

The authors have revised the manuscript accordingly. I am also satisfied with their response to my comments. I have no further questions.

Reviewer #3 (Remarks to the Author):

The authors carefully addressed all the comments provided by me and the other reviewers and, in my opinion, the quality of the manuscript has markedly improved. I only have a few additional comments.

1. Abstract: I like the changes but I would highlight the fact that the results here can inform cooling strategies/policies on the best type of “green spaces” as the last sentence (lines 24-27) is a bit vague. In other words, the key message is that trees are more “efficient” than treeless green spaces but such efficiency varies across climatic regions and average/extreme weather conditions – this is the key finding with policy relevance that should be highlighted.

>>>> We understand that the last sentence may be a bit vague. We reformulated it keeping in mind that the abstract length should not increase and that reviewer 1 expresses concerns regarding the very direct policy implications (see also reviewer 1, comment 2). The reformulated sentence highlights the multifaceted relationships that have to be considered, which we consider a major contribution in assisting policy-makers:

“By revealing continental-scale patterns in the effect of trees and treeless green spaces on urban LST our results highlight the importance of considering and further investigating the climate-dependent effectiveness of heat mitigation measures in cities.”

2. Actually, a recent study has come to similar conclusions (Paschalis et al. 2021) so a discussion/comparison would be useful (e.g. in the introduction or discussion section). In this context, I agree with the comment of reviewer 1 (see comment 1.5 in the rebuttal) that the focus on trees vs no-trees is somewhat limited compared to NDVI or LAI-based approaches but might be more “useful” for policy decisions (even if, as discussed by the authors, the definition of “trees” is also limited, as a line of street trees is certainly different from a dense urban forest). Given the potential interest of the topic for a wide audience, a few clarifications along these lines might be beneficial.

>>>> Thank you for mentioning the study of Paschalis et al. 2021, which is very interesting and indeed has some overlap with our results. We included it in the discussion. We also tried to be clearer that we think that it is not only different LAIs that matter for their influence on LSTs and air temperatures. For example, two tree species (or areas with different amounts of trees) with the same LAI may have a different root depth and hence may be able to extract more or less water, which may be particular relevant during hot/dry extremes. In addition, they may have a different behavior in terms of stomatal conductance. In addition, surface roughness and albedo may vary for different vegetation types with similar LAIs. Hence, while LAI may be able to explain a large part of the different effects of trees and treeless green spaces on temperatures, there are additional effects that can be captured by differentiating the vegetation types.

“... There are substantial temperature differences between tree-covered areas and green-spaces and between rural forests and rural pastures in several parts of Europe. As a recent study shows, such LST differences are related to high rates of evapotranspiration being linked to high leaf area indices (LAIs) of

tree-covered areas (Paschalis et al., 2021) and hence the study concludes, in accordance with our results, that not only the amount of green spaces, but also the type of vegetation exerts a strong control on LSTs and surface urban heat islands. Differences in evapotranspiration between vegetation types may not only be related to varying LAIs, but also to additional physiological and biological characteristics of different vegetation types and their control on evapotranspiration and surface roughness (Teuling et al., 2010, Yosef et al., 2018, Burakowski et al., 2018, Duveiller et al., 2018). For example, trees are associated with a larger root depth (Schenk and Jackson, 2002) that allows higher exploitation of soil moisture, sustaining larger evapotranspiration rates when the upper soil layers are dry (Yosef et al., 2018). ...”

3. Introduction: I appreciate the additions made by the authors on the physical mechanisms leading to cooling by trees. However, I would suggest to partly restructure this section to improve the logical flow of thoughts. For example, the fact that vegetation “can either increase or decrease temperature” is repeated in Line 68 and 73, the effect of water stress is mentioned in Line 66 and 78, etc.

>>>> We understand that there can be the impression that some aspects in this part of the introduction are redundant. However, while the first mentioning of “increase/decrease of temperatures and water stress” is in the paragraph that highlights seasonality and effects of climate extremes, the second paragraph mentioning these two points (i.e. increase/decrease of temperatures and water stress) highlights differences in the environmental conditions between within the city vs. outside of the city. We would like to keep these two paragraphs separated, because they explore different aspects of how the effects of vegetation on temperatures are influenced in space and time.

In general, the writing/language still needs some improvements (see below for a few suggestions).

4. Line 37: consider rephrasing as “potential cooling effect of trees in ...”

>>>> We rephrased: “...that the cooling effect of an increased amount of urban vegetation in tropical cities will be limited...”

5. Line 42: consider rephrasing as “as demonstrated by several studies”

>>>> rephrased: “... which has been shown in...”

6. Line 48: remove “e.g.” or rephrase (“e.g. see ref. 7”). Check this throughout the manuscript.

>>>> e.g. was removed and the sentence “rearranged”

Line 52: “mainly on ...”

>>>> Done. Thanks for spotting this.

Lines 55-56: the effect of temperature on stomatal conductance is true for vegetation in general, not just trees – please clarify

>>>> The paragraph (containing lines 55-56) is strongly focused on trees. Thus, we mention the stomatal conductance of trees. We don't think that readers will get the impression that we think stomatal conductance is not relevant for vegetation in general.

Line 63: not sure "ET" has been defined before

>>>> Thanks for spotting this. After deleting parts of the paragraph it has not been defined before.

Line 99: "LST observations"

>>>> Done. (Landsat removed)

Line 117: "regions of France, ..."

>>>> Done.

Line 239: "treeless urban green spaces"?

>>>> Yes. Done.

Lines 252-255: what is the reasoning behind this statement? Clarify

>>>> We removed this sentence.

Line 298: I think "surrounding", rather than "neighbourhood", is more appropriate here

>>>> Yes. Changed to surrounding.

Line 300: thanks for clarifying this – however, the sentence is still unclear to me. The fact that urban fabric has little or no ET is indeed expected but it is not a "minor effect" on LST differences – such differences are high precisely because ET is low over urban surfaces. I suggest rephrasing this.

>>>> We removed the sentence. It did indeed not add much new information.

Lines 320-324: see Paschalis et al. (2021) for a discussion on urban-rural differences in LAI and SIF.

>>>> We think that Paschalis et al. (2021) raise some important points, which is why we included the study in the discussion (see response to comment 2).

Line 335: I think MODIS is mentioned here for the first time – either explain upfront or write something like "the MODIS ET product used here..."

>>>> Done.

Line 388: this sentence sounds unclear to me

>>>> We removed the sentence since we dedicated a whole paragraph to discuss LST/air temperature relationships.

Lines 400-401: unclear how LCZs are related to the work here – is this a suggestion for improvement of the method? Is it an alternative? Please elaborate (...ok, now I see that the sentence is different in the rebuttal, please correct the main text).

>>>> The main text has been corrected.

Line 427: socio-economic factors? What does this mean?

>>>> We removed the term “socio-economic factors” which was intended to be used to indicate the many impacts of heat in cities on human/social behavior and related to economic damage (e.g. damages to roads), but was admittedly not very precise in this context.

Line 429: this statement is “out of the blue”. I suggest rephrasing as “In conclusion, we presented”

>>>> Yes, it indeed comes out of the blue. We added “In conclusion,...”

Line 431: UHI data are generally obtained from LST and LULC data which are then aggregated at the city-scale – so not sure I agree with this sentence.

>>>> We removed the second part of this sentence. This could be a very interesting discussion. What we want to emphasize is that there can be advantages of comparing the LSTs of (rather) precisely defined LULC categories. The quantification of the (surface) urban heat island is in our opinion often (but as mentioned not always) based on “mixed” LULC categories. For example, the category urban/city is often still including certain fractions of different types of green spaces. Important knowledge gains are then made by explaining the (S)UHIs in different regions based on several factors that also include the fraction of vegetation inside of the city. However, this approach may be less “direct” than when estimating temperatures for each LULC type separately (of course acknowledging that each LULC type may still have certain “mixed properties” even at very high resolution).

Figure 2: are the results in panel d also illustrated in panel c? If yes, I would remove panel d or simplify/clarify. Also, what do colours in panel d (boxplots) indicate?

>>>> Thank you for this suggestion. Panel c and d have been merged and the coloring removed.

Figure 3: clarify in the legend that urban green spaces are treeless

>>>> Done.

Figure S1: what is the legend for the colours in panel a? The colorbar in panel b seems to show a different temperature range

>>>> The colors in the boxplot have now been removed. They only had “illustrative purpose”.

Figure S3: I would use the same temperature limits/colorbar for the top and bottom panels

>>>> Yes that makes sense. Done.

Figure S4: for consistency, urban fabric should be indicated by UF

>>>> Yes, thanks. Done.

Figures S15-S20 are not discussed in the text – I suggest adding a supplementary section motivating/explaining these analyses. Actually, all these supplementary figures/results should be mentioned also in the main text (not sure whether this is a journal requirement but it is certainly useful for the reader).

>>>> The figures are now either referenced in the main text (including the methods section). In addition, we created a short supplementary section for explanation regarding Figure S 15 and S 16.

References

Paschalis, A., Chakraborty, T. C., Fatichi, S., Meili, N., & Manoli, G. (2021). Urban forests as main regulator of the evaporative cooling effect in cities. *AGU Advances*, 2(2), e2020AV000303.

References (in the response)

- BURAKOWSKI, E., TAWFIK, A., OUIMETTE, A., LEPINE, L., NOVICK, K., OLLINGER, S., ZARZYCKI, C. & BONAN, G. 2018. The role of surface roughness, albedo, and Bowen ratio on ecosystem energy balance in the Eastern United States. *Agricultural and Forest Meteorology*, 249, 367-376.
- DUVEILLER, G., HOOKER, J. & CESCATTI, A. 2018. The mark of vegetation change on Earth's surface energy balance. *Nature Communications*, 9, 679.
- LI, D., LIAO, W., RIGDEN, A. J., LIU, X., WANG, D., MALYSHEV, S. & SHEVLIAKOVA, E. 2019. Urban heat island: Aerodynamics or imperviousness? *Science Advances*, 5, eaau4299.
- MANOLI, G., FATICHI, S., SCHLÄPFER, M., YU, K., CROWTHER, T. W., MEILI, N., BURLANDO, P., KATUL, G. G. & BOU-ZEID, E. 2019. Magnitude of urban heat islands largely explained by climate and population. *Nature*, 573, 55-60.
- MARTILLI, A., KRAYENHOFF, E. S. & NAZARIAN, N. 2020. Is the Urban Heat Island intensity relevant for heat mitigation studies? *Urban Climate*, 31, 100541.
- OKE, T. R., MILLS, G., CHRISTEN, A. & VOOGT, J. A. 2017. *Urban Climates*, Cambridge, Cambridge University Press.
- PASCHALIS, A., CHAKRABORTY, T., FATICHI, S., MEILI, N. & MANOLI, G. 2021. Urban Forests as Main Regulator of the Evaporative Cooling Effect in Cities. *AGU Advances*, 2, e2020AV000303.
- SCHENK, H. J. & JACKSON, R. B. 2002. THE GLOBAL BIOGEOGRAPHY OF ROOTS. *Ecological Monographs*, 72, 311-328.
- TEULING, A. J., SENEVIRATNE, S. I., STOCKLI, R., REICHSTEIN, M., MOORS, E., CIAIS, P., LUYSSAERT, S., VAN DEN HURK, B., AMMANN, C., BERNHOFER, C., DELLWIK, E., GIANELLE, D., GIELEN, B., GRUNWALD, T., KLUMPP, K., MONTAGNANI, L., MOUREAUX, C., SOTTOCORNOLA, M. & WOHLFAHRT, G. 2010. Contrasting response of European forest and grassland energy exchange to heatwaves. *Nature Geoscience*, 3, 722-727.
- WANG, D., LIANG, S., HE, T., YU, Y., SCHAAF, C. & WANG, Z. 2015. Estimating daily mean land surface albedo from MODIS data. *Journal of Geophysical Research: Atmospheres*, 120, 4825-4841.
- WANG, L., HUANG, M. & LI, D. 2020. Where Are White Roofs More Effective in Cooling the Surface? *Geophysical Research Letters*, 47, e2020GL087853.
- YOSEF, G., WALKO, R., AVISAR, R., TATARINOV, F., ROTENBERG, E. & YAKIR, D. 2018. Large-scale semi-arid afforestation can enhance precipitation and carbon sequestration potential. *Scientific Reports*, 8, 996.

REVIEWERS' COMMENTS

Reviewer #1 (Remarks to the Author):

Within the limitations of the study, which the authors have now more clearly stated, I am now satisfied with the revised manuscript.

Reviewer #3 (Remarks to the Author):

The authors included in the manuscript all the requested changes. The study is of good quality and, apart for a few minor suggestions (see below), I do not have any additional comments.

Line 40: why focusing on tropical cities here when the study is about Europe? I would keep the discussion more general (e.g. dry vs wet climates).

Lines 52-84: the discussion here is valid for both trees and treeless vegetation – maybe add some clarifications explaining why we expect different cooling effects by trees and grasses/shrubs.

Lines 75: “surrounding land”? Do you mean “local microclimate” here?

Fig 1: DeltaT should be DeltaT_UF-UT, right? Also, the value of DeltaT is negative in the legend/colorbar but positive in the y axis/stacked bars – explain or revise.

Lines 325-326: which “data on the biophysical processes”? Maybe add a few examples.

Lines 338-341: as stated in the previous lines, this could be due to soil moisture availability and not just VPD – maybe rephrase.

Line 394: remove “e.g.”

Lines 410-412: I would rephrase as: “... to comprehensively analyse ... that aims at separating ...”. Also, remove the quotation mark (”) at the end of the sentence.

Lines 452-462, remove quotation marks (“...”)

Fig. S1: what is DeltaT here? UT-UF? Check this everywhere (e.g. Fig. S8-9)

Fig. S10 and Line 1106: I would define R2, e.g. “the coefficient of determination, R^2 ”

Fig. S13: what is Q_{JJA}^* in panel c?

We thank all reviewers for their time and effort and their very helpful and constructive comments. We would particularly like to thank reviewer 3 for their very detailed comments and the tremendous help in improving this manuscript. In the following, we list the reviewers' comments in black and our reply in blue.

Reviewer #1 (Remarks to the Author):

Within the limitations of the study, which the authors have now more clearly stated, I am now satisfied with the revised manuscript.

Reviewer #3 (Remarks to the Author):

The authors included in the manuscript all the requested changes. The study is of good quality and, apart for a few minor suggestions (see below), I do not have any additional comments.

Line 40: why focusing on tropical cities here when the study is about Europe? I would keep the discussion more general (e.g. dry vs wet climates).

We modified: "...the cooling effect of an increased amount of urban vegetation in tropical cities will be limited and generally differs between wet and dry climates ²"

Lines 52-84: the discussion here is valid for both trees and treeless vegetation – maybe add some clarifications explaining why we expect different cooling effects by trees and grasses/shrubs.

Yes, we agree that part of this introduction is also valid for treeless vegetation. However, since the discussion on cooling effects of trees and treeless green spaces (and their differences) is already quite extensively covered (line 292-314), we would not add any further explanations to the introduction.

Lines 75: "surrounding land"? Do you mean "local microclimate" here?

Thank you for spotting this. Indeed, we mean the influence of local- and micro-scale climatic conditions (defined by the surrounding land). We reformulated: "The potential of trees to reduce temperatures via transpiration is influenced by local- and micro-scale climatic conditions..."

Fig 1: DeltaT should be DeltaT_UF-UT, right? Also, the value of DeltaT is negative in the legend/colorbar but positive in the y axis/stacked bars – explain or revise.

Yes, thank you, we included UT-UF and for the stacked bars now also include the negative sign.

Lines 325-326: which “data on the biophysical processes”? Maybe add a few examples.

We modified: “..., it will be crucial to generate high spatial resolution data on the biophysical processes within cities including, e.g., estimates of sensible and latent heat fluxes³³.”

Lines 338-341: as stated in the previous lines, this could be due to soil moisture availability and not just VPD – maybe rephrase.

We agree and clarified: “Since the cooling of urban trees during hot extremes shifts north and increases over the British Isles, Scandinavia and parts of Mid-Europe/Alps, we assume that higher VPD in combination with sufficient soil moisture availability causes an increase in transpiration in those regions. The decreased cooling during hot extremes in the Mediterranean and Turkey indicates that increased VPD will not lead to a further increase in transpiration in southern regions due to limited soil moisture.”

Line 394: remove “e.g.”

Removed.

Lines 410-412: I would rephrase as: “... to comprehensively analyse ... that aims at separating ...”. Also, remove the quotation mark (”) at the end of the sentence.

Yes, that improves the sentence. We modified according to the suggestion.

Lines 452-462, remove quotation marks (“...”)

Removed.

Fig. S1: what is DeltaT here? UT-UF? Check this everywhere (e.g. Fig. S8-9)

We now specified deltaT not only in the figure captions but directly in Figure S8 and S9. In addition, we specified it in Figure S11, Figure S 16 and Figure S 18.

Fig. S10 and Line 1106: I would define R2, e.g. “the coefficient of determination, R²”

Done.

Fig. S13: what is Q_JJA_* in panel c?

These are the different temperature quantiles for which we have calculated the fraction of observed days. We now clarified this in the caption of Figure S13. We also changed the figure by replacing the fraction with the absolute number of days, which is more consistent with panels a and b.